

# Tropospheric Ozone Seasonal and Long-term Variability as seen by lidar and
surface measurements at the JPL-Table Mountain Facility, California
Maria Jose Granados-Muñoz[1] and Thierry Leblanc[1]
1 Jet Propulsion Laboratory, California Institute of Technology, Wrightwood, CA, USA
Corresponding author: mamunoz@jpl.nasa.gov
Keywords: NDACC, lidar, ozone, troposphere, surface ozone, TOLNet, long-term,
tropopause folds, UTLS



Abstract
A combined surface and tropospheric ozone climatology and interannual variability study
was performed for the first time using co-located ozone photometer measurements (2013-2015)
and tropospheric ozone differential absorption lidar measurements (2000-2015) at the Jet
Propulsion Laboratory Table Mountain Facility (TMF, elev. 2285 m), in California.
The surface time-series were investigated both in terms of seasonal and diurnal variability.
The observed surface ozone is typical of high-elevation remote-sites, with small amplitude of the
seasonal and diurnal cycles, and high ozone values, compared to neighboring lower altitude
stations representative of urban boundary layer conditions. The ozone mixing ratio ranges from
45 ppbv in the winter morning hours to 65 ppbv in the spring and summer afternoon hours. At
the time of the lidar measurements (early night), the seasonal cycle observed at the surface is
similar to that observed by lidar between 3.5 km and 9 km.
Above 9 km, the local tropopause height variation with time and season impacts significantly
the ozone lidar observations. The frequent tropopause folds found in the vicinity of TMF (27%
of the time, mostly in winter and spring) produce a dual vertical structure in ozone within the
fold layer, characterized by higher-than-average values in the bottom half of the fold (12-14 km),
and lower-than-averaged values in the top half of the fold (14-18 km). This structure is consistent
with the expected origin of the air parcels within the fold, i.e., mid-latitude stratospheric air
folding down below the upper tropospheric sub-tropical air.
No significant signature of interannual variability could be observed on the 2000-2015
deseasonalized lidar time-series, with only a statistically non-significant positive anomaly during
the years 2003-2007. Our trend analysis reveals however a statistically significant positive trend
of 0.31 ppbv.year$^{-1}$ in the free troposphere for the period 2000-2015.
A classification of the air parcels sampled by lidar was made at 1-km interval between 5 km
and 14 km altitude, using 8-days backward trajectories (HYSPLIT). Our classification revealed
the large influence of the Pacific Ocean, with air parcels of low ozone content (50-65 ppbv), and
significant influence of the stratosphere leading to ozone values of 65-80 ppbv down to 8-9 km.



In summer, enhanced ozone values (70 ppbv) were found in air parcels originating from Central
America, probably owed to the enhanced thunderstorm activity during the North American
Monsoon. No outstanding influence from Asia was identified.





## 1. Introduction

Ozone is an important constituent in the troposphere, impacting climate, chemistry, and air quality (The Royal Society, 2008). As a greenhouse gas (Forster et al., 2007), it contributes to the Earth's global warming with an estimate radiative forcing of $0.40 \pm 0.20$ W·m$^{-2}$ (IPCC 2013). It is one of the main oxidants in the troposphere (Monks, 2005), and, in high concentrations, it can cause problems in human health and vegetation (World Health Organization, 2003). Tropospheric ozone can be directly emitted to the troposphere, but it is primarily formed as a secondary pollutant in chemicals reactions involving ozone precursors such as methane, CO, NOx, VOCs or PANs. An additional source of ozone in the troposphere is the downward transport from the stratosphere, where ozone is much more abundant (Levy et al., 1985). At high elevation sites such as the Jet Propulsion Laboratory Table Mountain Facility in Southern California (TMF hereafter), the effect of the boundary layer is very small, and ozone variability is expected to be driven by transport processes from the stratosphere or horizontal transport within the troposphere (Cui et al., 2009; Naja et al., 2003; Trickl et al., 2010).

Several studies show that background ozone levels have increased significantly since preindustrial times (Mickley et al., 2001; Parrish et al., 2012; Staehelin et al., 1994; Volz and Kley, 1988) and these levels continued rising in the last decades in both Hemispheres (Derwent et al., 2007; Jaffe et al., 2004; Lee and Akimoto, 1998; Naja and Akimoto, 2004; Oltmans et al., 2006; Parrish et al., 2012; Simmonds et al., 2004; Tanimoto et al., 2009; Zbinden et al., 2006; Lelieveld et al., 2004). Nevertheless, after air quality regulations were implemented in the 1970s, the increasing trend has slowed down or is even reverted in regions such as the Eastern U.S. and Europe (Cooper et al., 2012, 2014; Granier et al., 2011). The situation is not the same for emerging economies such as Asia, where emissions are increasing with the corresponding increase in ozone levels (Dufour et al., 2010; Gao et al., 2005; Strode et al., 2015; Tie et al., 2009; Wang et al., 2006).

In most cases, variability and trend studies have revealed very large ozone variability with time, location and altitude (Cooper et al., 2014). This variability is mostly due to the large heterogeneity and variability of the ozone sources themselves, the different chemical processes affecting the formation and depletion of tropospheric ozone and its variable lifetime in the troposphere. Ozone atmospheric lifetime goes from a few hours in polluted boundary layer to



several weeks in the free troposphere, allowing it to travel over distances of intercontinental scale (Stevenson et al., 2006; Young et al., 2013). In order to obtain statistically significant results and be able to assess tropospheric ozone interannual variability and trends, a large long-term monitoring dataset with global coverage is required, which has not yet been achieved considering the current observation capabilities.

Long-term records of tropospheric ozone are available since the 1950s (Feister and Warmbt, 1987; Parrish et al., 2012), but it is not until the 1970s that the number of ozone monitoring stations became significant (Cooper et al., 2014 and references therein). Currently, a considerable number of ozone monitoring sites are operating as part of regional networks or international programs (e.g. World Meteorological Observation Global Atmosphere Watch WMO/GAW, Acid Deposition Monitoring Network in East Asia EANET, Clean Air Status and Trends Network CASTNET, etc.). In addition to these ground-based networks, tropospheric ozone measurements from satellite (TOMS, TES, OMI, etc.) or aircraft (MOZAIC/IAGOS) platforms have been successfully implemented. Nevertheless, most of the tropospheric ozone measurements are still only surface or column-integrated measurements whilst the number of them with information on the vertical coordinate is very scarce. Until today, mainly ozonesonde profiles have been used to provide altitude-resolved ozone variability information in the troposphere (Logan, 1994; Logan et al., 1999; Naja and Akimoto, 2004; Oltmans et al., 1998, 2006, 2013), but the somewhat elevated cost of an ozonesonde launch has kept the sampling interval to one profile per week (or less) for a given location. Differential Absorption Lidar (DIAL) systems, which started to be used to measure tropospheric ozone in the late 1970s (Bufton et al., 1979), complement the ozonesonde records, providing higher temporal resolution thanks to their inherent operational configuration (from minutes to days of continuous measurements). Today, tropospheric ozone lidars are still very scarce, but the implementation of observation networks such as the international Network for the Detection of Atmospheric Composition Change (NDACC, http://www.ndsc.ncep.noaa.gov), and more recently the North American-based Tropospheric Ozone Lidar Network (TOLNet, http://www-air.larc.nasa.gov/missions/TOLNet) allows for new capabilities that can contribute to the understanding of processes affecting tropospheric ozone variability, and to satellite and model validation and improvement.





As part of NDACC and TOLNet, a tropospheric ozone DIAL system located at TMF has
been operating since 1999. In this study, an analysis of 16 years of lidar profiles measured at the
station is presented together with the analysis of the surface ozone measurements available at the
site since 2013. The objective is to provide the first-ever published study of tropospheric ozone
variability above TMF using both the surface and lidar datasets. The work presented here is
particularly valuable due to the rising interest in the detection of long-term trends in the Western
United States (U.S.) and the scarcity of long-term measurements of ozone vertical profiles in this
region. The high terrain elevation and the deep planetary boundary layer of the intermountain
Western U.S. region facilitate inflow of polluted air masses originating in the Asian boundary
layer and ozone-rich stratospheric air down to the surface, thus highly influencing air quality in
the region (Brown-Steiner and Hess, 2011; Cooper et al., 2004; Langford et al., 2012; Liang et
al., 2004; Lin et al., 2012a, 2012b; Stohl, 2002). After a brief description of the instrumentation
and datasets (Section 2), an analysis of the seasonal and interannual variability of tropospheric
ozone above TMF for the period 2000-2015 will be presented in section 3. The study includes a
characterization of the air parcels sampled by lidar by identification of the source regions based
on backward trajectories analysis. A summary and discussion are provided in Section 4.
2.  Instrumentation
2.1 Tropospheric ozone lidar
TMF is located in the San Gabriel Mountains, in Southern California (34.4◦ N, 117.7◦ W), at an
elevation of 2285 m above sea level. Two differential absorption lidars (DIAL) and one Raman
lidar have been operating at the facility during nighttime typically four times per week, two
hours per night, contributing stratospheric ozone, temperature, tropospheric ozone, and water
vapor measurements to NDACC for several decades now. The original design in the mid-1990s
of the tropospheric ozone DIAL was optimized for tropospheric ozone and aerosol measurements
(McDermid, 1991). The system was later re-designed to provide exclusively tropospheric ozone
profiles (McDermid et al., 2002). The emitter uses a quadrupled Nd:YAG laser emitting two
beams at 266 nm. One beam is sent into a Raman cell filled with Deuterium to shift the
wavelength to 289 nm, the other beam is sent into another cell filled with Hydrogen to shift the
wavelength to 299 nm. The two beams are then expanded five times and transmitted into the
atmosphere. The light elastically backscattered in the troposphere (3-20 km) is collected by



several telescopes comprising mirrors of diameters varying from 91 cm diameter to 5 cm diameter, thus accommodating for the large signal dynamic range implied when collecting light from this close range. A total of three pairs of 289/299 nm channels is thus used to retrieve ozone using the DIAL technique, each pair corresponding to a different intensity range and the retrieved ozone profiles from all pairs combined together ultimately covering the entire troposphere (3-18 km). As part of the retrieval process, the upper range of the ozone profile is further extended to about 25 km by applying the DIAL technique on the 299 nm high intensity channel of the tropospheric ozone lidar and the 355 nm low-intensity channel of the co-located water vapor Raman lidar (Leblanc et al., 2012).

The instrument temporal sampling can be set to any value from a few seconds to several hours, depending on the science or validation need. The vertical sampling can be set to any multiple of 7.5 m, again depending on the science or validation need. For the routine measurements contributing to NDACC over the period 1999-2015 and used for the present work, the standard settings have typically ranged between 5-min and 20-min for temporal sampling, and between 7.5 m and 75 m for the vertical sampling. Profiles routinely archived at NDACC are averaged over 2-hours, with an effective vertical resolution varying from 150-m to 3 km, depending on altitude. These temporal and vertical resolution settings yield a standard uncertainty of 7-14% throughout the profile. The system operates routinely at nighttime, but daytime measurements with reduced signal-to-noise ratio are occasionally performed in special circumstances such as process studies, and aircraft or satellite validation. The total number of routine 2-hour ozone profiles used in this study and archived at NDACC for the period 2000-2015 is included in Table 1.

The TMF ozone lidar measurements have been regularly validated using simultaneous and co-located Electrochemical Concentration Cell (ECC) sonde measurements (Komhyr, 1969; Smit et al., 2007). In the troposphere the precision of the ozonesonde measurement is around 3-5%. TMF has ozonesonde launch capability since 2005 and 32 coincident profiles were obtained over the period 2005-2013. Results from the lidar and the ECC comparison are included in Figure 1. Figure 1c reveals that the deviations do not present significant changes with time, which is an indicator of the system stability despite the multiple upgrades made over this time period. In most cases, differences were within ±15% for the complete analyzed period. Note that





a non-negligible fraction of the differences is due to geophysical variability. The measurement
geometry of the lidar and ozonesonde are radically different: 2-hour averaged, single location for
lidar, and horizontally-drifting instantaneous measurements for the ozonesonde.
2.2. Surface ozone measurements
Continuous surface ozone measurements have been performed at TMF since 2013 with a
UV photometric ozone analyzer (Model 49i from Thermo Fisher Scientific, US). The operation
principle is based on the absorption of UV light at 254 nm by the ozone molecules. The
instrument collects in-situ air samples at 2 meter above ground taken from an undisturbed
forested environment adjacent to the lidar building. It provides ozone mixing ratio values at 1-
minute time intervals with a lower detection limit of 1 ppbv.
3. Results

3.1.Surface ozone variability
Figure 2a shows the surface ozone seasonal cycle at TMF and nearby stations from the
California Air Resources Board (ARB) air quality network for the period 2013-2015. The
seasonal cycle at TMF comprises a maximum in spring and summer and a minimum in winter,
consistent with the ARB stations shown, as well as other stations in the US West Coast (e.g.
Schnell et al., 2015). Nonetheless, the seasonal cycle obtained at TMF from the hourly samples
(left plot) presents larger ozone values and lower variability throughout the year compared to the
other ARB stations. The mean surface value for the complete period at TMF is 55 ppbv, whereas
the seasonal values are 57, 57, 52 and 45 ppbv in spring (March-April-May), summer (June-July-
August), fall (September-October-November) and winter (December-January-February)
respectively. These values are in good agreement with those obtained from surface
measurements at high elevation sites in the Northern Hemisphere and reported in the review by
Cooper et al., (2014). When using the 8hMDA (8-h maximum daily average, right plot), larger
seasonal cycle amplitudes are observed, especially at stations affected by anthropogenic
pollution such as Crestline or San Bernardino. These polluted stations present larger values in
summer than those recorded at high-elevation remote stations like Joshua Tree or TMF. The
mean 8hMDA at TMF is 58 ppbv and the seasonal averages are 62, 66, 57 and 49 for spring,



summer, fall and winter respectively. The observed low seasonal variability is typical of high
elevation remote sites with low urban influence (Brodin et al., 2010). A similar behavior can be
observed at the Phelan, Joshua Tree or the Mojave National Preserve stations, all sites being at
high elevation with low or negligible urban influence. In Figure 2a a secondary minimum is
observed at TMF and most of the ARB nearby stations in July-August, followed by a secondary
maximum in fall.

In Figure 2a a clear combined effect of the altitude and proximity to anthropogenic pollution

sources on the ozone levels is observed. In general, higher ozone levels and lower variability are
observed at higher altitudes. The lowest altitude Pico Rivera instrument measures the lowest
ozone levels, and the highest-altitude TMF instrument measures the highest ozone levels
throughout the year when considering the hourly sampled dataset. A mean difference of ~30
ppbv is observed for a 2-km altitude difference. The magnitude of this positive ozone vertical
gradient depends on the distance from anthropogenic pollution sources. The effect of pollution is
clearer on the 8hMDA data, where high-elevation stations, yet more likely to be affected by
pollution such as Crestline or Victorville, present a larger seasonal cycle amplitude associated
with lower ozone levels in winter and higher levels in summer. A similar impact of the interplay
between urban influence and high-elevation was previously reported by Brodin et al., (2010).

The difference between the seasonal cycle retrieved from the 1-hour averaged data and the

8hMDA can be easily explained from the differences in the daily cycles at the different stations.
The mean surface ozone diurnal cycle at TMF and nearby ARB stations is shown in Figure 2b
for the four seasons. Minimum values are observed at nighttime, whereas maxima appear in late
afternoon. As for the seasonal cycle, the daily cycle at TMF, Joshua Tree, Mojave National
Preserve and Phelan stations exhibit low variability compared to the other stations located at
lower altitude and more affected by urban pollution. On average, daily values are larger at high
elevation remote sites as TMF or Joshua Tree. However, the afternoon maximum is larger at
polluted stations such as Crestline, especially in the summer season. In addition, the maximum at
TMF and the ARB stations of Joshua Tree and Mojave National Preserve occurs later than at the
other stations. The difference in timing is likely due to the different chemical species involved in
the ozone formation and depletion processes due to the low influence of anthropogenic pollution
(Brodin et al., 2010; Gallardo et al., 2000; Naja et al., 2003). In winter, a minimum is observed at



TMF in the afternoon instead of the maximum observed at the other stations. This difference in
diurnal pattern has been observed at other remote or high-elevation sites and has been attributed
to the shorter day length and the lack of ozone precursors compared to urban sites. The resulting
daytime photochemical ozone formation is insufficient to produce an ozone diurnal variation
maximizing in the afternoon (Brodin et al., 2010; Gallardo et al., 2000; Naja et al., 2003;
Oltmans and Komhyr, 1986; Pochanart et al., 1999; Tsutsumi and Matsueda, 2000).

3.2.Tropospheric ozone variability

The red curve in Figure 3a (left plot) shows the average tropospheric ozone profile obtained by
the TMF lidar for the period 2000-2015. The cyan horizontal bars show the corresponding
standard deviation at 1-km interval. The red dot at the bottom of the profile shows the 2013-2015
mean surface ozone obtained from the data acquired simultaneously to the lidar measurements.
The lidar system can provide information from around 1.3 km (200 meters since 2013) above the
surface up to 25 km, covering the whole troposphere and the lower stratosphere. The average
mixing ratio value in the mid-troposphere is 55 ppbv. Above 8 km, the ozone mixing ratio
increases, reaching values above 1 ppmv at 16 km.

The seasonally averaged profiles are shown in Figure 3b. They show larger values in

spring and summer in the troposphere, whereas in the stratosphere maximum values are observed
in winter and spring. Within the troposphere, below 9 km, the seasonally-averaged profiles show
average values of 62, 60, 51 and 50 ppbv in spring, summer, fall and winter respectively. These
values are in good agreement with the average ozone concentrations (50-70 ppbv) obtained in
previous studies (Thompson et al., 2007; Zhang et al., 2010) above the western U.S. In the
altitude range 9-16 km (UTLS) a much larger variability in ozone is observed, as indicated by
the large standard deviation (left plot) and the differences between the seasonally-averaged
profiles (right plot). This large variability results from the horizontal and vertical displacement of
the tropopause above the site, causing the lidar to sound either the ozone-rich lowermost
stratosphere or the ozone-poor sub-tropical upper troposphere for a given altitude.

The 2D color contours of Figure 4 show the composite (2000-2015) monthly mean ozone

climatology measured by lidar (main panel, 4-20 km). A similar 2D color contour representation
was used just below the main panel to represent the composite (2013-2015) monthly mean





surface ozone. The climatological tropopause height at TMF is also included in the main panel
(blue dotted line), with mean values ranging between 12 and 15 km. As discussed previously in
this paper, the tropopause height variability is the main cause of the larger standard deviation
observed in Figure 3a in this region. Between the surface and 9 km, a very consistent seasonal
pattern is observed, with maximum values in April-May and minimum values in winter. The
spring-summer maximum in the free-troposphere has been consistently observed at other stations
in Europe and North America and is attributed to photochemical production (Law et al., 2000;
Petetin et al., 2015; Zbinden et al., 2006). Above 9 km, the seasonal maximum is observed
earlier, i.e., in March and April between 10 and 12 km and February and March at higher
altitudes, consistent with the transition towards a dynamically-driven lower stratospheric regime.
At these altitudes, the ozone minimum is also displaced earlier in the year (August-October),
which is consistent with the findings of Rao et al. (2003) above Europe.

The TMF surface and lidar data are found to be very consistent, both in terms of seasonal

cycle phase and amplitude, and in term of absolute mixing ratio values. The mean value obtained
from the lidar measurements in the troposphere is very similar to the mean value obtained from
the surface measurements (around 55 ppbv).

The consistency between the lidar and the surface data was found not only for the

seasonal cycle obtained from the monthly averaged values, but also for the complete time series.
The degree of correlation between the lidar measurements at the lowest point and the surface
measurements was investigated. As mentioned before, the lidar cannot measure all the way down
to the surface. The first valid measurement occurs at around 3.5-4 km depending on the time
period. Therefore, the layer from 4 to 6 km is considered as the lower lidar layer.  A correlation
coefficient of R=0.34 was found between the lidar data in the layer from 4 to 6 km and the
surface data. The correlation increases (R=0.44) if we consider a 3-hour time lag between the 4-6
km layer and the surface. After removing outliers corresponding to ozone values higher (or
lower) than the average plus (minus) one standard deviation either at the surface or at 4-6 km, the
correlation increases up to 0.69 for the simultaneous data and up to 0.79 for the 3-hour time-
shift.

3.3. Interannual variability and trends




The 2000-2015 time-series of the deseasonalized ozone mixing ratio is shown in Figure

5. Anomalies, expressed in percent, were calculated by subtracting the climatological ozone
monthly mean profiles computed for the period 2000-2015 to the measured lidar profiles. Large
ozone variability with time is clearly observed, highlighting the difficulty to identify trends and
patterns. No clear mode of interannual variability is observed for the analyzed period here.
However, positive anomalies seem to predominate throughout the troposphere during the period
2003-2007, especially below 7 km. On average, ozone mixing ratio values in the lower
troposphere were 5 ppbv larger in 2003-2007 than during the entire period 2000-2015.

Following a procedure similar to that described in Cooper et al. (2012), a trend analysis

was performed at different altitude levels (Tables 2 and 3 and Figure 6). Figure 6 shows the time
series of the median, 95[th] and 5[th] percentile values, obtained every year between 2000 and 2015
for different layers and different seasons using the lidar profiles measured at TMF. In order to
obtain the trends, linear fits (shown in Figure 6) of the median, 95[th] and 5[th] percentiles were
performed independently using the least squares method. The ozone rate of change in ppbv.year[-1]
was determined from the slope of the linear fit. To assess the significance of the trends, the F-
statistic test was used, with the p-Value as an indicator of the statistical significance. For p-
Values lower than 0.1, trends were assumed statistically significant, with a confidence level
larger than 90%.

The calculated trends were found to depend on altitude and season. Table 2 contains the

ozone rate change expressed in ppbv.year[-1] (and %.year[-1]) for the different layers and seasons for
the median, 5[th] and 95[th] percentiles. The corresponding standard errors and p-Values are included
in Table 3. Statistically significant trends are marked in bold font. The layer corresponding to the
upper troposphere (7-10 km) shows a statistically significant ozone increase of 0.31 ppbv.year[-1]
(0.57%·year[-1]) for the median values and 0.55 ppbv.year[-1] (0.54%.year[-1]) for the 95[th] percentile,
indicating that both the background and the high intensity events ozone levels were increasing
(Cooper et al., 2012, 2014). A similar increase in the free troposphere and in the western US was
reported by Cooper et al. (2012) for the period 1990-2010 for both the median and 95[th]
percentiles.

Now looking at each season separately, a significant positive trend was found in the

upper troposphere (7-10 km) for both spring and summer, with an ozone increasing rate of 0.71
and 0.58 ppbv.year[-1] respectively (or 1.10 and 0.98%·year[-1]), and an ozone decrease of -0.43



ppbv.year$^{-1}$ (-0.87%·year$^{-1}$) during winter. Statistically significant trends were also found in the
lower troposphere (4-7 km) during winter for the median and 5$^{th}$ percentile values with an ozone
decrease of -0.36 ppbv.year$^{-1}$ and -0.59 ppbv.year$^{-1}$ respectively (-0.72 and -1.53%·year$^{-1}$
respectively) (Table 2). No significant trend was observed near the tropopause (12-16 km),
whereas a significant negative trend of -8.79 ppbv.year$^{-1}$ (-1.39%·year$^{-1}$) for the median and -
5.80 ppbv.year$^{-1}$ (-1.26%·year$^{-1}$) for the 5$^{th}$ percentile in fall was observed in the lower
stratosphere (17-19 km).
3.4. Characterization of the air masses sounded by the TMF tropospheric ozone lidar
In an attempt to characterize the air parcels sounded by lidar above TMF based on their travel
history, 8-day backward trajectories ending at TMF between 5 and 14 km altitude were
computed using the HYSPLIT4 model (Draxler and Rolph, 2003),
http://www.arl.noaa.gov/ready/hysplit4.html). The NCAR/NCEP Reanalysis Pressure level data
were used as meteorological input (Kalnay and Kanamitsu, 1996) in HYSPLIT4. These data,
available since 1948, provide 4-times-daily meteorological information at 17 pressure levels
between the 1000 and 10 hPa and 2.5x2.5 degrees horizontal resolution. Several studies (Harris
et al., 2005; Stohl, 1998) provided a wide range of uncertainty estimates along the trajectories.
The more recent study by Engström and Magnusson, (2009) indicate values of 354-400 km
before 4 days and 600 km after.
Our trajectory analysis comprises two steps. First, the 8-day backward trajectories computed
by HYSPLIT and ending at different altitude levels were grouped using the HYSPLIT clustering
tool (Draxler et al., 2009) in order to identify the most significant paths followed by the air
masses arriving over the station. Based on the results of this preliminary analysis, five main
regions were identified: the stratosphere, the Asian boundary layer (ABL), the free-troposphere
above Asia (AFT), Central America, and the Pacific Ocean. Once these geographical areas were
identified, we performed a classification of the air parcels according to the criteria described
next.
An air parcel was classified as "stratospheric" if the 8-day backward trajectory intercepted
the tropopause and resided at least 2 days above the local tropopause. The tropopause height
information comes from the global tropopause height data derived once a day by the NOAA



Physical Sciences Division (http://www.esrl.noaa.gov/psd) from the same NCAR/NCEP
Reanalysis database used as input to HYSPLIT4. Computations are based on the World
Meteorological Organization (WMO, 1957) definition, that is, the lowest height at which the
temperature lapse rate becomes lower than 2 K·km$^{-1}$, provided that along 2 km above this height
the average lapse is also lower than 2 K·km$^{-1}$. In addition, the NOAA computations do not allow
tropopause heights at pressure levels larger than 450 hPa and smaller than 85 hPa.
Next, the air parcels that were not classified as "stratosphere" were then classified as "Asian
boundary layer" (ABL) for trajectories comprising a minimum residence time of 2 days within
the area labelled "ABL" in Figure 7, and below an altitude of 3 km. Next, air parcels with
trajectories comprising a minimum residence time of 2 days within the area labelled "AFT" in
Figure 7, and above 3 km altitude were classified as "Asian Free-troposphere" (AFT).
The air parcels not classified as "stratosphere", "ABL", or "AFT" were then classified as
"Central America" if the corresponding trajectories' residence time over the "Central America"
region depicted in Figure 7 was found to be at least 2 days. Finally, the remaining parcels were
classified as "Pacific Ocean" when the residence time above the corresponding region (see figure
7) reached or exceeded 2 days.
The classification of the air parcels took place sequentially, which means that each
category is exclusive of the others. The classification was made for each of the four seasons
separately in order to account for the seasonal changes in synoptic circulation. Examples of the
corresponding classified back-trajectories are shown in Figure 8. The number and frequency of
occurrences of each air parcel category for all seasons is compiled in Table 4. A monthly
distribution of these occurrences is shown in Figure 9. With the selection criteria we have set, a
very low number of parcels classified "ABL" were found. Air parcels dominantly originate in the
Pacific Ocean below 10 km, with almost equal influence throughout the year. Increasing
influence of the stratosphere is observed at upper levels, with values ranging between 2 to 78%
or higher during winter and spring. This result agrees well with previous studies in the Western
US (Sprenger, 2003; Stohl, 2003). A statistically significant Central American influence was
identified in summer with a frequency of occurrence varying between 11% and 3%, decreasing
with altitude. The Central America influence coincides with the establishment of the North



American Monsoon circulation from July to September and which affects Central America and the Southwestern US.

Composite ozone profiles and statistical parameters were estimated for each category of air parcel and for altitudes between 4.5 and 13.5 km at 1-km altitude intervals. Figure 10 shows the ozone mixing ratio mean (open circles), median (red bars), $25^{th}$ and $75^{th}$ percentiles (blue bars) at 9 km altitude for each of the identified categories and season. The number of occurrences for each category is mentioned between parentheses. The ozone statistics obtained when a low number of occurrences was found should be ignored (e.g., Central America in Spring, or ABL for most seasons). Figure 11 shows, for each season, the composite ozone profiles constructed from the ozone mixing ratio median values found for a particular category at a given altitude. In order to keep the most statistically significant results, composite values computed using less than 5% of the total number of samples for a given season were not plotted, leaving out certain sections of the composite profiles, and in the case of ABL, leaving out the entire profile.

Not surprisingly, the analysis reveals that larger ozone mixing ratio values were observed when the air masses were classified as "stratospheric" regardless of altitude and season (65-85 ppbv below 9 km). For this category, large ozone variability was found, as indicated by the $25^{th}$ and $75^{th}$ percentiles in Figure 10. As altitude increases, the influence of the stratosphere is more important, exceeding 30% above 12 km, resulting in higher ozone mixing ratio values (red curves in Figure 11).

Conversely, low ozone mixing ratio values (50-65 ppbv below 9 km) were consistently associated with the air parcels classified as "Pacific Ocean" (cyan curves). This region can be considered as a source of 'background ozone', since no anthropogenic source is expected to affect the local ozone budget. Slightly higher ozone content (50-70 ppbv higher) is systematically found for air parcels classified as "AFT", but the number of occurrences remains too small to provide any meaningful interpretation. No conclusion can be made for air parcels classified as "ABL" due to the very low number of occurrences found.

During summer, ozone values of about 70 ppbv were found at 9 km altitude for the 85 air parcels classified as "Central America" (yellow curve). The corresponding values for the 277 air



parcels classified as "Pacific Ocean" are about 55 ppbv, which is 15 ppbv lower. This difference
possibly points out to the lightning-induced enhancement of ozone within the more frequent
occurrence of thunderstorms during the North American summer monsoon.

3.5. The influence of tropopause folds on the TMF tropospheric ozone record

In the previous section, a large variability in the composite ozone content was found for the air
parcels classified as "stratospheric". In the current section, we provide at least one clear
explanation for this large variability. Tropopause folds are found primarily in the vicinity of the
subtropical jets, in the 20º-50º latitude range. They typically consist of three-dimensional folds of
the virtual surface separating air masses of tropospheric characteristics (weakly stratified, moist,
low ozone concentration, etc.) and those of stratospheric characteristics (highly stratified, dry,
high ozone concentration, etc.). Tropopause folds can result in the transport of large amounts of
stratospheric ozone into the troposphere, reaching in some cases the planetary boundary layer
and enhancing ozone amounts even at the surface (Chung and Dann, 1985; Langford et al., 2012;
Lefohn et al., 2012; Lin et al., 2012a). They are considered one of the main mechanisms of
stratosphere-to-troposphere exchange and have been widely studied in the past (e.g. Bonasoni
and Evangelisti, 2000; Danielsen and Mohnen, 1977; Lefohn et al., 2011; Vaughan et al., 1994;
Yates et al., 2013). Due to the location of TMF, the upper troposphere above the site is
frequently impacted by tropopause folds. The MERRA reanalysis (1-km vertical resolution, 1 x
1.25 degrees horizontal resolution) were used in this study to identify the presence of double
tropopauses above the station. A comparison between the MERRA temperature profiles and the
temperature profiles measured by the radiosondes launched at TMF was performed in order to
evaluate the performance of MERRA above TMF. The comparison (not shown) reveals excellent
agreement, with average relative differences of 2% or less from the surface up to 25 km. The
heights of double tropopauses were computed following a methodology similar to that proposed
in Chen et al., (2011). The first (lower) tropopause is identified according to the WMO
definition, as explained earlier. A second (upper) tropopause is identified above the WMO
tropopause if the temperature lapse rate increases over 3 K·km$^{-1}$ within at least 1 km, and its
height is determined once again by the WMO criterion.

Using this methodology, we found that 27% of the TMF tropospheric ozone lidar profiles

were measured in the presence of double tropopauses. Figure 12 shows the number of cases with





double tropopauses above TMF distributed per months, with the number of days with tropopause
folds being normalized to the total number of measurements every month (compiled in Table 1).
As we can see, the presence of double tropopauses was especially frequent during winter and
spring, which coincides with the higher frequency of stratospheric air masses arriving at TMF
estimated by the backward trajectories analysis (Figure 9). The altitude of detected single
tropopauses is found around 13 km in winter and spring, and 16-17 km in summer and fall
(Figure 13a-d). When a double tropopause is identified, the altitude of the lower tropopause
ranges between 8 and 15 km, with the distribution peak centered around 12-13 km (Figure 13e-
h), and the second tropopause is detected typically around 17-18 km (Figure 13i-l).
Figure 14 shows the average of all tropospheric ozone lidar profiles measured in winter
(blue curves) and spring (red curves) in the presence of a double tropopause (solid curves), and
in the presence of a single tropopause (dashed curves). The right panel (Figure 14b) is simply a
lower tropospheric-zoomed version of the left panel (Figure 14a). Only winter and spring are
shown because they are the seasons most affected by double tropopause cases as previously
stated. In the presence of double tropopauses a clear dual vertical structure in ozone is observed,
with higher ozone values between 12 and 14 km and lower mixing ratio values between 14 and
18 km. In the case of deep stratospheric intrusions, ozone-rich stratospheric air masses embedded
in the lower half of the fold can reach lower altitudes, and occasionally the planetary boundary
layer mixing down to the surface (Chung and Dann, 1985; Langford et al., 2012, 2015; Lefohn et
al., 2012; Lin et al., 2012a), leading to an ozone increase in the lower troposphere (Figure 14b).
In our case, the mean increase is around 2 ppbv below 6 km for both spring and winter. Note the
relative magnitude of the ozone anomalies in the lower and upper halves of the fold is different
for spring and winter. A detailed investigation, beyond the scope of the present work, is needed
to assess the actual significance of this difference.
4.  Discussion
The present study allowed the characterization of the full tropospheric ozone profile from the
ground to the stratosphere, and is particularly valuable in the context of a notoriously sparse
horizontal coverage for this type of vertically resolved measurements in a region affected by
transboundary ozone inflow and stratospheric intrusions.



The combined analysis of the surface measurements and the simultaneous lidar profiles
reveals high consistency between the ozone at the surface and in the free troposphere. This
consistency may point out the fact that the TMF surface measurements are representative of the
lower part of the free troposphere (i.e., below 7 km), at least during the nighttime lidar
measurements. Additional daytime lidar measurements will be performed in 2016 to assess
whether such consistency also exists at other times of the day, especially in the afternoon.
The analysis of the long-term lidar time-series (16 years covered) shows no significant
signatures of interannual variability, as previously discussed. More importantly, no obvious
signature of ENSO or the QBO could be identified, which is inconsistent with the recent findings
of Lin et al., (2015) or Neu et al., (2014), who have linked tropospheric ozone variability in the
Northern Hemisphere to El Niño/ La Niña events, and the QBO, through the variations of
stratospheric/tropospheric ozone fluxes. However, this inconsistency might not be so surprising
if we take into account the obvious difference in measurement sampling (one single location in
the Western U.S. as opposed to global observations).
Nevertheless, our analysis reveals statistically-significant trends for selected seasons and
altitudes. Specifically, a positive trend of 0.31 ppbv.year$^{-1}$ (ozone annual median) was found in
the upper troposphere (7-10 km). This positive trend is more pronounced in spring and summer
(0.71 and 0.58 ppbv.year$^{-1}$ respectively), while a negative trend (-0.43 ppbv.year$^{-1}$) was found in
winter. The positive trend obtained here in spring for the median values is larger than the trend
obtained by Cooper et al. (2012) for the free troposphere in 1995-2011 (0.41 ppbv.year$^{-1}$), and
even larger than the trend obtained by Lin et al. (2015b) using model data (0.37 ppbv.year$^{-1}$
during 1995-2012). This disagreement could be due to differences in sampling, as concluded in
Lin et al. (2015b). Nonetheless, Figure 6 shows larger ozone median (and 5[th] and 95[th] percentile)
values at 7-10 km in 2013-2015 than in preceding years, with this period not being included in
the previous studies. A lower ozone increasing rate in 2000-2012 above TMF (0.43 ppbv.year$^{-1}$)
suggests that that the ozone rate of change has increased in the last years, but a more
comprehensive study with regional coverage would be necessary to confirm this change.
Regarding winter season, a positive trend was obtained on a regional scale in Cooper et al.,
(2012), but certain sites in the western U.S. showed a negative trend, as in our case. This
negative trend indicates a decrease in the background ozone values. During winter months, a





smaller influence of transboundary ozone transport is expected at low altitudes above TMF and
the decrease in background ozone during these months could be associated with the decrease in
domestic anthropogenic emissions.
The springtime positive trend estimates reported in the Western US oppose ozone
decrease in the Eastern part. These results indicate that the two-decade-long efforts to implement
regulations to control air quality and anthropogenic emissions in the U.S. have led to a clear
decrease in ozone levels in the Eastern U.S., but not in the Western U.S. (e.g. Copper et al.,
2012; 2014). This different regional behavior has been attributed to the inflow of elevated ozone,
mainly from East Asia, and to the increasing contribution of stratospheric intrusions (Cooper et
al., 2010; Jacob et al., 1999; Parrish et al., 2009; Reidmiller et al., 2009). But again, differences
in sampling can impact significantly the interpretation of our trend estimates. As pointed out by
Lin et al. (2015b), further coordination efforts at both global and regional scales are necessary in
order to reduce biases introduced by inhomogeneity in sampling. As part of these efforts, an
extended analysis based on the origin of the air masses sampled by the TMF lidar is under way,
with the ultimate objective to filter synoptic noise out, and better quantify the ENSO and QBO
signals and the residual trends. As a prerequisite to such study, a preliminary characterization of
the air masses sounded by the TMF lidar was performed and presented here.
Backward trajectories analysis reveals that, not surprisingly, parcels identified as
"stratosphere" contain the highest ozone mixing ratios, and parcels classified as "Pacific Ocean"
contain the lowest ozone concentrations, which can be considered as 'background conditions'.
Despite influence of Asian pollution in the ozone levels in the Western US has been detected in
previous studies (e.g. Zhang et al., 2008; Lin et al., 2012; Langford et al., 2015), no outstanding
signature from the Asian boundary layer could be identified due to the low number occurrences
associated with this category in our case. Nevertheless, air parcels classified as "Asian free-
troposphere" seemed to contain systematically more ozone than those classified as "Pacific
Ocean", especially below 9 km. A refined classification, probably requiring the use of chemistry-
transport models, is needed to assess whether the Asian boundary layer or the Asian free-
troposphere have a detectable impact on the ozone content measured above TMF.
In summer, air masses coming from Central America were frequently detected. The
ozone mixing ratio values measured in this case were clearly above the climatological mean,





with values up to 15 ppbv larger than those associated with the Pacific Ocean region. Previous
studies (Cooper et al., 2009), have observed enhanced ozone values associated with the North
American Monsoon, mainly due to ozone production associated with lightning (Choi et al., 2009;
Cooper et al., 2009). However, this feature was observed in the Eastern U.S. Because of the
synoptic conditions during the monsoon, the Western U.S. is not as much influenced and no
significant ozone increase was reported (Barth et al., 2012; Cooper et al., 2009). Nevertheless,
Cooper et al., (2009) reported higher modeled lightning-induced $NO_x$ concentrations at TMF
than at other western locations, which would be consistent with our findings. Further
investigation, including a detailed history of the meteorological conditions along the trajectories,
is needed to confirm this correlation.

As part of the characterization of the TMF lidar-sampled air masses, the impact of double

tropopauses was assessed. Between 2000 and 2015, the frequency of occurrence of double
tropopauses above TMF was found to be around 27%. This high frequency, especially observed
in winter and spring, was expected considering the latitude of TMF, i.e., near the subtropical jet,
where frequent tropopause folds occur. More interestingly, a clear, dual vertical structure in
ozone was observed in the presence of a double tropopause (Figure 14). The double tropopause
is expected to result from a tropopause fold in the layer between the two identified tropopauses.
The dual ozone structure observed by lidar coincides with the expected location of the fold, and
consists of systematically higher-than-average mixing ratios in the lower half of the fold (12-14
km), and lower-than-average mixing ratios in the upper half of the fold (14-18 km). This dual
structure is consistent with the expected origin of the air masses within a tropopause fold.
Stratospheric air, richer in ozone, is measured within the lower half of the fold, while
tropospheric ozone-poor air is measured within the upper half of the fold. In addition,
statistically significant higher-than-average mixing ratios (+2 ppbv) are observed in the lower
troposphere (4-10 km) in the presence of double-tropopauses. This increase is consistent with
previous reports of the importance of the stratosphere as an ozone source in the lower
troposphere (Cooper and Stohl, 2005; Langford et al., 2012; Lefohn et al., 2011; Trickl et al.,
2011), with a 25 to 50% contribution to the tropospheric budget (Davies and Schuepbach, 1994;
Ladstätter-Weißenmayer et al., 2004; Roelofs and Lelieveld, 1997; Stevenson et al., 2006).





560  Further investigation is now underway, with the objective of better identifying all

561 signatures of interannual variability and trends. This long-term variability investigation not only

562 will include an air mass classification based on geographical area, but will also take into account

563 the history of the air parcels in the context of nearby tropopause folds. Altogether, this air parcel

564 characterization, used in conjunction with regional chemistry-climate or chemistry-transport

565 model runs, should unveil signatures hidden in the current lidar record, and eventually shed new

566 light on the origin of the reported past, current, and future tropospheric ozone trends over the

567 Western U.S.

568 5. Concluding remarks

569  Combined ozone photometer surface measurements (2013-2015) and tropospheric ozone

570 DIAL profiles (2000-2015) at the JPL-Table Mountain Facility were presented for the first time.

571 The high ozone values and low interannual and diurnal variability measured at the surface,

572 typical of high elevation remote sites with no influence of urban pollution, constitute a good

573 indicator of background ozone conditions over the Southwestern US.

574  The 16-year tropospheric ozone lidar time-series is one of the longest lidar records

575 available and is a valuable dataset for trend analysis in the Western US, where the number of

576 long-term observations with high vertical resolution in the troposphere is very scarce. A

577 statistically significant positive trend was observed in the upper troposphere, in agreement with

578 previous studies. This ozone increase points out to the influence of long-range transport and/or a

579 change in stratospheric influence, since ozone precursor emissions have been decreasing in the

580 US over the past two decades.

581  Ozone vertical distribution above TMF is affected by the frequent occurrence of

582 tropopause folds. A dual vertical structure in ozone within the fold layer was clearly observed,

583 characterized by above-average values in the bottom half of the fold (12-14 km), and below-

584 averaged values in the top half of the fold (14-18 km). Above-average ozone values were also

585 observed near the surface (+2 ppbv) on days with a tropopause fold. The high frequency of

586 tropopause folds observed above the site is not surprising given Table Mountain's position in the

587 vicinity of the subtropical jet. A detailed and systematic analysis of the ozone vertical structure,

588 and its variability in relation to the proximity of the subtropical jet is in preparation. This



analysis, which will utilize a large amount of model data together with the lidar observations,
should shed some light on a possible correlation between the observed upper tropospheric ozone
positive trends and the impact of the tropopause folds, either as a result of an increase in
frequency of the stratospheric intrusions, or as a result of the mid-latitude stratospheric ozone
recovery observed since the early 2000s (WMO, 2014).

ACKNOWLEDGEMENTS
The work described in this paper was carried out at the Jet Propulsion Laboratory,
California Institute of Technology, under a Caltech Postdoctoral Fellowship sponsored by the
NASA Tropospheric Chemistry Program. Support for the lidar, surface, and ozonesonde
measurements was provided by the NASA Upper Atmosphere Research Program. The authors
would like to thank M. Brewer, T. Grigsby, J. Howe, and members of the JPL Lidar Team who
assisted in the collection of the data used here. The authors gratefully acknowledge the NOAA
Air Resources Laboratory (ARL) for the provision of the HYSPLIT transport and dispersion
model and/or READY website (http://www.ready.noaa.gov) and the NCEP/NCAR Reanalysis
team for the data used in this publication. We would also like to thank Dr. Susan Strahan and the
MERRA Reanalysis team for providing the data used in this study and to acknowledge the
California Air Resources Board for providing the surface ozone data.

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





**Tables:**

Table 1. Number of measurements, by month and years, performed at TMF with the tropospheric
ozone DIAL system. N/A indicates data not available at the time of the study

|  | Jan | Feb | Mar | Apr | May | Jun | Jul | Aug | Sep | Oct | Nov | Dec | Total |
|---|---|---|---|---|---|---|---|---|---|---|---|---|---|
| **2000** | 4 | 2 | 6 | 4 | 11 | 12 | 7 | 10 | 8 | 1 | 4 | 2 | 65 |
| **2001** | 1 | 11 | 17 | 2 | 9 | 13 | 12 | 15 | 15 | 17 | 1 | 11 | 130 |
| **2002** | 6 | 10 | 6 | 4 | 0 | 10 | 11 | 1 | 6 | 16 | 6 | 10 | 93 |
| **2003** | 11 | 9 | 15 | 12 | 10 | 13 | 5 | 7 | 9 | 14 | 11 | 9 | 117 |
| **2004** | 9 | 8 | 15 | 14 | 12 | 6 | 12 | 13 | 11 | 10 | 9 | 8 | 130 |
| **2005** | 4 | 6 | 13 | 8 | 12 | 16 | 9 | 2 | 7 | 2 | 4 | 6 | 99 |
| **2006** | 11 | 9 | 6 | 8 | 14 | 5 | 2 | 12 | 12 | 20 | 11 | 9 | 106 |
| **2007** | 0 | 0 | 4 | 9 | 11 | 7 | 8 | 10 | 8 | 26 | 0 | 0 | 101 |
| **2008** | 7 | 11 | 8 | 13 | 9 | 4 | 11 | 10 | 6 | 11 | 7 | 11 | 100 |
| **2009** | 14 | 11 | 7 | 5 | 7 | 8 | 4 | 10 | 4 | 17 | 14 | 11 | 91 |
| **2010** | 0 | 0 | 3 | 8 | 0 | 7 | 4 | 1 | 4 | 5 | 0 | 0 | 44 |
| **2011** | 2 | 6 | 4 | 7 | 7 | 11 | 10 | 12 | 7 | 8 | 2 | 6 | 90 |
| **2012** | 0 | 9 | 9 | 1 | 10 | 13 | 3 | 2 | 5 | 8 | 0 | 9 | 69 |
| **2013** | 6 | 3 | 5 | 10 | 8 | 7 | 5 | 7 | 0 | 0 | 6 | 3 | 51 |
| **2014** | 9 | 2 | 5 | 10 | 13 | 16 | 15 | 11 | 15 | 15 | 14 | 6 | 131 |
| **2015** | 9 | 15 | 12 | 18 | 3 | 15 | 12 | N/A | N/A | N/A | N/A | N/A | 83 |
| **Total** | 93 | 112 | 135 | 133 | 136 | 162 | 130 | 123 | 117 | 170 | 103 | 86 | 1500 |






Table 2. Ozone mixing ratio trends for the median, 5[th] and 95[th] percentiles over the period 2000-2015 as shown in Figure 6 (see text
for details) in ppbv·year$^{-1}$ (%·year$^{-1}$). Statistically significant trends are marked in bold font

| | Ozone mixing ratio trends in ppbv/year (%/year) | | | | | | | | | | | | | | |
|---|---|---|---|---|---|---|---|---|---|---|---|---|---|---|---|
| | Year | | | Spring | | | Summer | | | Fall | | | Winter | | |
| | Med. | 5[th] P. | 95[th] P. | Med. | 5[th] P. | 95[th] P. | Med. | 5[th] P. | 95[th] P. | Med. | 5[th] P. | 95[th] P. | Med. | 5[th] P. | 95[th] P. |
| **17-19 km** | -0.49 (-0.05) | 0.25 (0.05) | -5.37 (-0.37) | -1.01 (-0.1) | 3.47 (0.68) | -5.89 (-0.41) | -2.93 (-0.44) | -3.25 (-0.63) | -0.13 (-0.01) | **-8.79 (-1.39)** | **-5.80 (-1.26)** | -6.58 (-0.68) | -0.12 (-0.01) | 1.37 (0.27) | -21.86 (-1.51) |
| **12-16 km** | 1.56 (1.01) | -0.01 (-0.01) | 2.52 (0.51) | 1.10 (0.50) | 0.58 (0.19) | 0.29 (0.05) | 0.08 (0.06) | 0.20 (0.30) | 0.19 (0.06) | -0.83 (-0.71) | -1.12 (-1.83) | -1.49 (-0.63) | 2.54 (1.31) | 0.51 (0.65) | 0.95 (0.18) |
| **7-10 km** | **0.31 (0.57)** | 0.01 (0.03) | **0.55 (0.54)** | **0.71 (1.10)** | 0.20 (0.49) | 4.31 (6.69) | **0.58 (0.98)** | 0.27 (0.90) | 1.01 (0.95) | -0.03 (-0.06) | -0.49 (1.62) | 0.18 (0.22) | **-0.43 (-0.87)** | -0.30 (-0.91) | -1.19 (-1.41) |
| **4-7 km** | -0.14 (-0.26) | -0.33 (-0.85) | 0.19 (0.17) | 0.12 (0.20) | -0.29 (-0.67) | 0.96 (1.17) | -0.14 (-0.24) | -0.03 (0.09) | -0.01 (-0.01) | -0.23 (0.45) | -0.82 (-2.33) | 0.26 (0.06) | **-0.36 (-0.72)** | **-0.59 (-1.53)** | 0.05 (0.08) |














Table 3. Standard errors in ppbv·year$^{-1}$ and p-Values associated to ozone mixing ratio trends for the median, 5$^{th}$ and 95$^{th}$ percentiles
included in Table 2 and shown in Figure 6. Data corresponding to statistically significant trends are marked in bold font

| Ozone mixing ratio trend standard errors in ppbv/year | | | | | | | | | | | | | | |
| --- | --- | --- | --- | --- | --- | --- | --- | --- | --- | --- | --- | --- | --- | --- |
| | Year | | | Spring | | | Summer | | | Fall | | | Winter | | |
| | Med. | 5$^{th}$ P. | 95$^{th}$ P. | Med. | 5$^{th}$ P. | 95$^{th}$ P. | Med. | 5$^{th}$ P. | 95$^{th}$ P. | Med. | 5$^{th}$ P. | 95$^{th}$ P. | Med. | 5$^{th}$ P. | 95$^{th}$ P. |
| **17-19 km** | 3.58 | 3.12 | 6.22 | 6.79 | 4.76 | 8.71 | 2.57 | 3.88 | 5.32 | **4.47** | **2.92** | 11.77 | 7.06 | 5.90 | 0.82 |
| **12-16 km** | 1.21 | 0.39 | 4.66 | 1.83 | 0.89 | 5.95 | 0.83 | 0.51 | 3.76 | 1.10 | 0.85 | 3.84 | 2.54 | 1.16 | 10.05 |
| **7-10 km** | **0.15** | 0.19 | **0.30** | **0.25** | 0.38 | 3.32 | **0.28** | 0.20 | 1.20 | 0.25 | 0.35 | 0.97 | **0.18** | 0.27 | 1.00 |
| **4-7 km** | 0.14 | 0.24 | 0.25 | 0.31 | 0.36 | 0.56 | 0.21 | 0.35 | 0.38 | 0.31 | 0.38 | 0.53 | **0.16** | **0.18** | 0.28 |


| p-Values | | | | | | | | | | | | | | |
| --- | --- | --- | --- | --- | --- | --- | --- | --- | --- | --- | --- | --- | --- | --- |
| | Year | | | Spring | | | Summer | | | Fall | | | Winter | | |
| | Med. | 5$^{th}$ P. | 95$^{th}$ P. | Med. | 5$^{th}$ P. | 95$^{th}$ P. | Med. | 5$^{th}$ P. | 95$^{th}$ P. | Med. | 5$^{th}$ P. | 95$^{th}$ P. | Med. | 5$^{th}$ P. | 95$^{th}$ |
| **17-19 km** | 0.89 | 0.94 | 0.40 | 0.88 | 0.48 | 0.51 | 0.27 | 0.41 | 0.98 | **0.07** | **0.07** | 0.60 | 0.99 | 0.82 | 0.17 |
| **12-16 km** | 0.22 | 0.98 | 0.60 | 0.55 | 0.52 | 0.96 | 0.92 | 0.71 | 0.96 | 0.47 | 0.21 | 0.70 | 0.29 | 0.67 | 0.92 |
| **7-10 km** | **0.06** | 0.94 | 0.09 | **0.01** | 0.60 | 0.22 | **0.05** | 0.19 | 0.41 | 0.91 | 0.18 | 0.86 | **0.03** | 0.28 | 0.25 |
| **4-7 km** | 0.33 | 0.19 | 0.44 | 0.70 | 0.44 | 0.11 | 0.52 | 0.92 | 0.98 | 0.47 | 0.05 | 0.63 | **0.04** | **4.10$^{-3}$** | 0.85 |





Table 4. Number of air parcels ending at TMF during lidar measurements over the period 2000-2015, classified as "Stratosphere", "Central America, "ABL","AFT" and "Pacific Ocean" (see text for details)

|  | Strat | Cent. Am | ABL | AFT | Pac |
|---|---|---|---|---|---|
| 14 km | 1150 (78%) | 47 (3%) | 0 (0%) | 5 (0%) | 270 (18%) |
| 13 km | 853 (58%) | 69 (5%) | 1 (0%) | 20 (1%) | 519 (35%) |
| 12 km | 515 (35%) | 86 (6%) | 4 (0%) | 61 (4%) | 798 (54%) |
| 11 km | 321 (22%) | 89 (6%) | 16 (1%) | 51 (3%) | 985 (67%) |
| 10 km | 199 (13%) | 94 (6%) | 17 (1%) | 93 (6%) | 1058 (71%) |
| 9 km | 118 (8%) | 98 (7%) | 18 (1%) | 92 (6%) | 1146 (77%) |
| 8 km | 70 (5%) | 103 (7%) | 19 (1%) | 82 (6%) | 1197 (91%) |
| 7 km | 62 (4%) | 110 (7%) | 19 (1%) | 84 (6%) | 1194 (81%) |
| 6 km | 24 (2%) | 133 (9%) | 11 (1%) | 67 (5%) | 1226 (83%) |
| 5 km | 31 (2%) | 161 (11%) | 12 (1%) | 87 (6%) | 1169 (79%) |



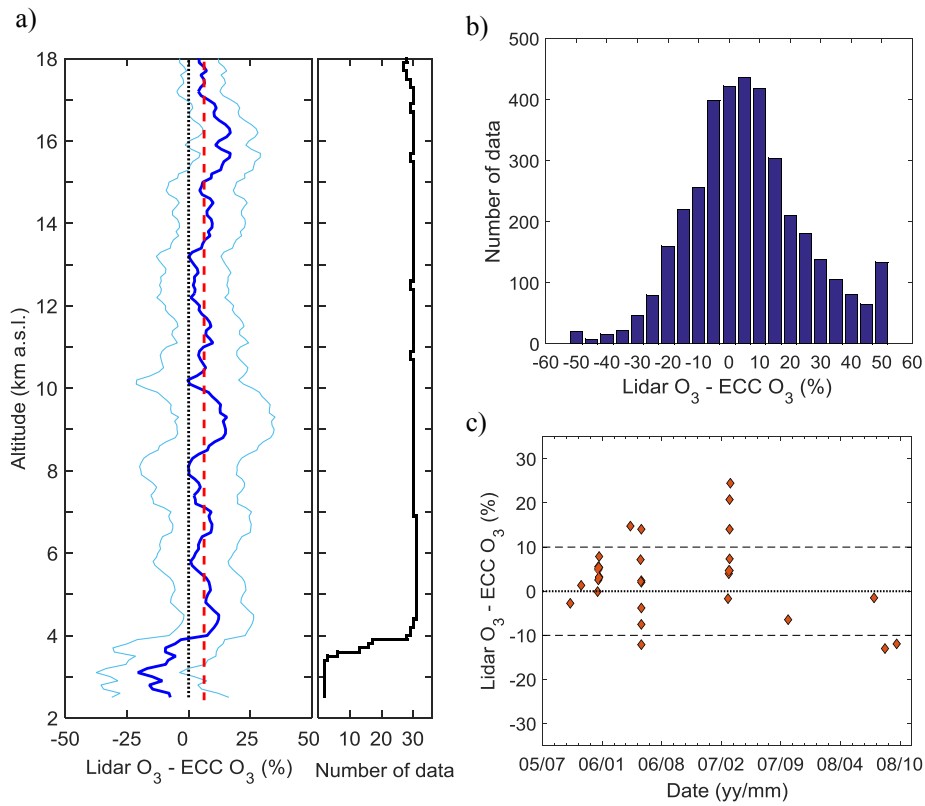

Figure 1. a) Profile of the mean relative difference between the lidar and the ECC ozone number density for the 32 simultaneous measurements (dark blue). Lidar uncertainty (light blue) and mean relative difference obtained between 4 and 16 km (red dotted line) are superimposed. The black solid curve shows the number of data points at each altitude. b) Histogram of the difference between the lidar and the ECC ozone number density. c) Column-averaged (below 8 km) difference between the lidar and the ECC sonde for each coincidence





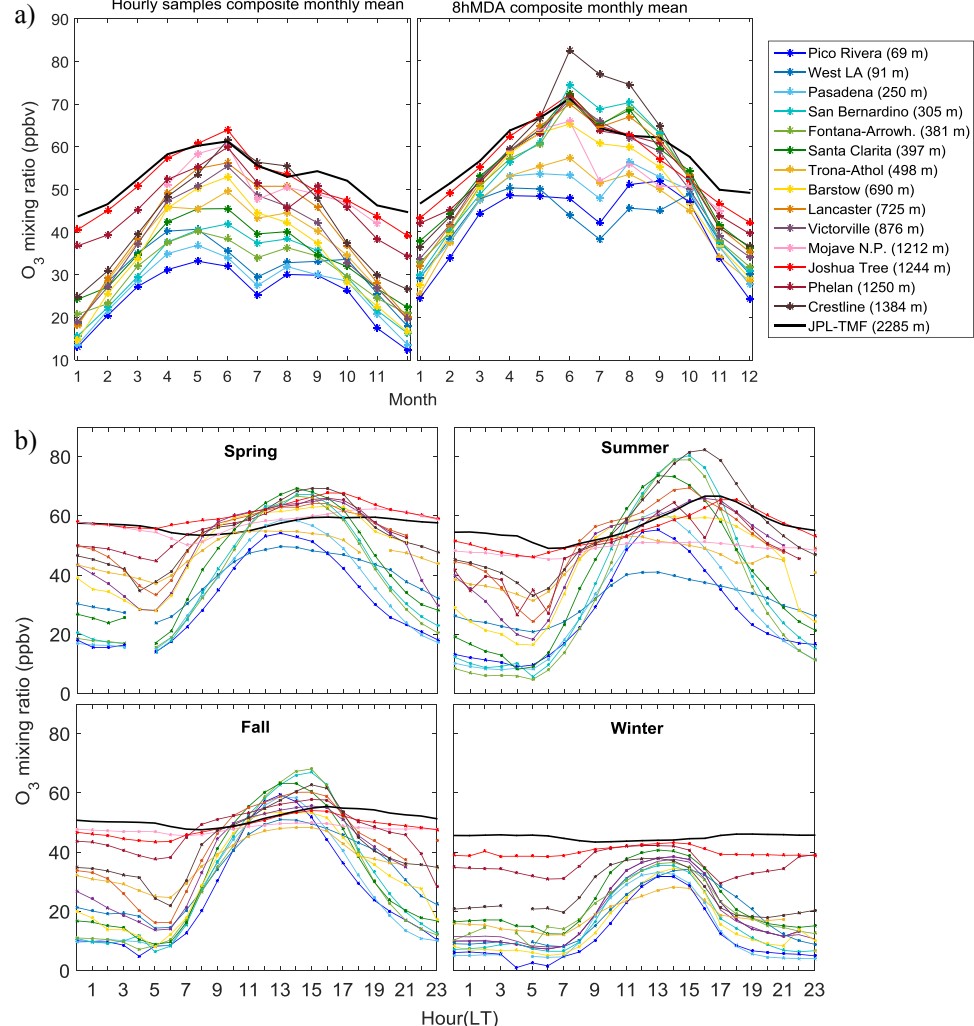

Figure 2. a) Composite monthly mean surface ozone at TMF and nearby ARB stations obtained from hourly samples (left) and 8hMDA values (right) for the period 2013-2015. b) Composite mean ozone daily cycle at TMF and nearby ARB stations for the four seasons for the period 2013-2015





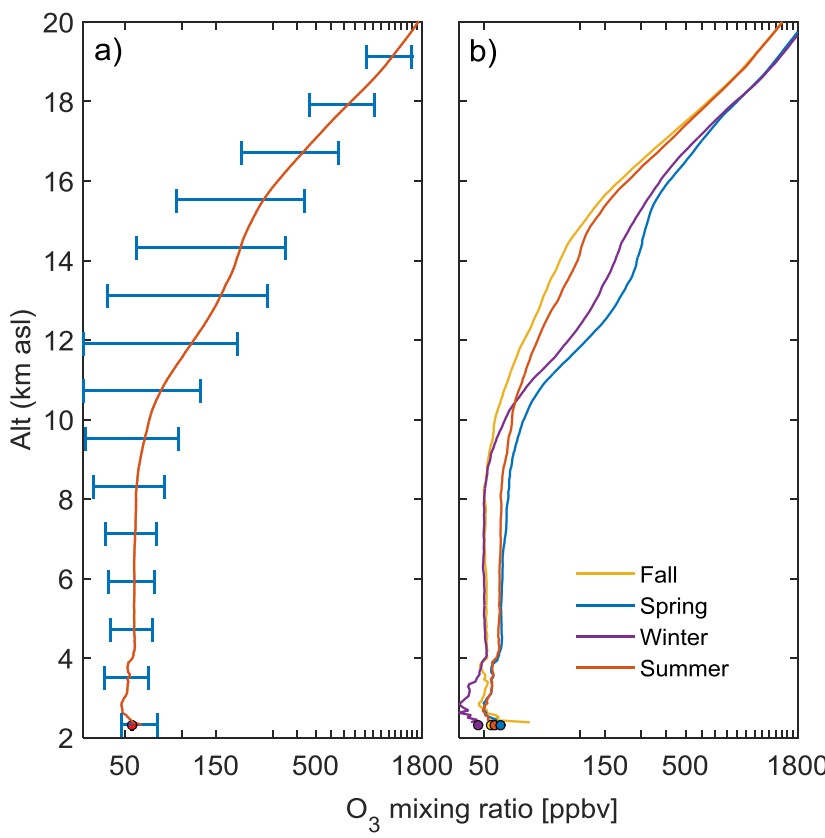

Figure 3. a) Ozone mixing ratio climatological average (2000-2015) computed from the TMF lidar measurements (red curve). The cyan horizontal bars indicate the standard deviation at intervals of 1-km. The red dot at the bottom indicates the mean surface ozone mixing ratio (2013-2015) measured simultaneously with lidar. b) Seasonally-averaged ozone mixing ratio profiles for spring (MAM), summer (JJA), fall (SON) and winter (DJF). The dots at the bottom indicate the corresponding surface ozone seasonal averages




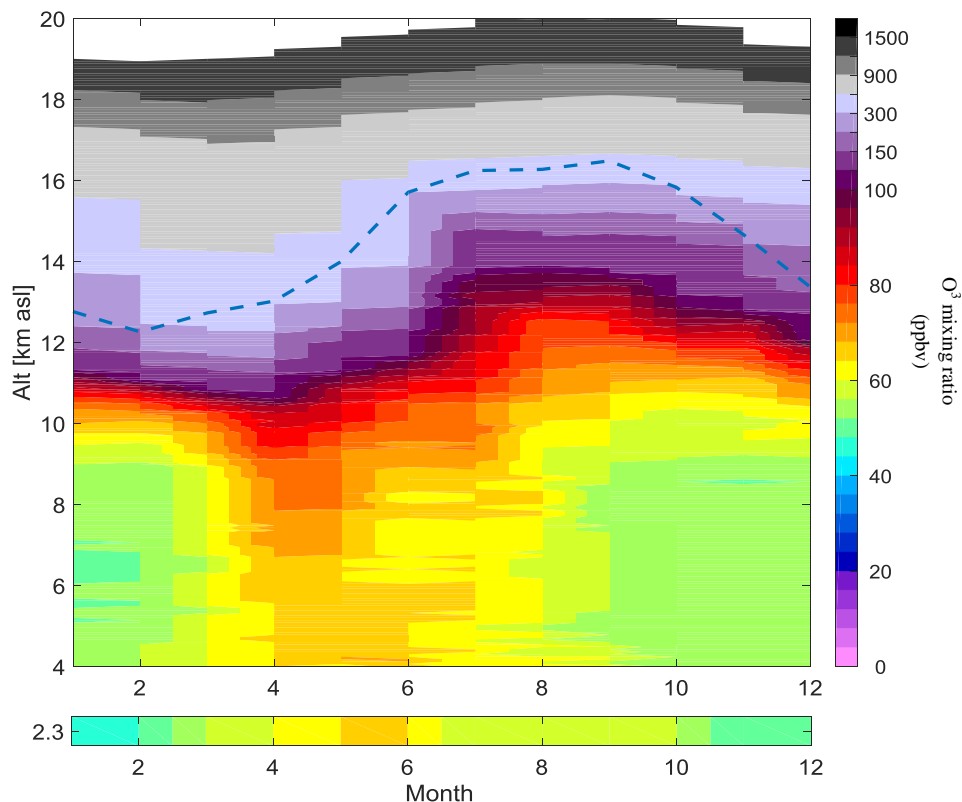

Figure 4. Composite monthly mean ozone mixing ratio (2000-2015) computed from the TMF lidar measurements. The dashed line indicates the climatological tropopause above the site (WMO definition). Bottom strip: Composite monthly mean ozone mixing ratio (2000-2015) from the surface measurements





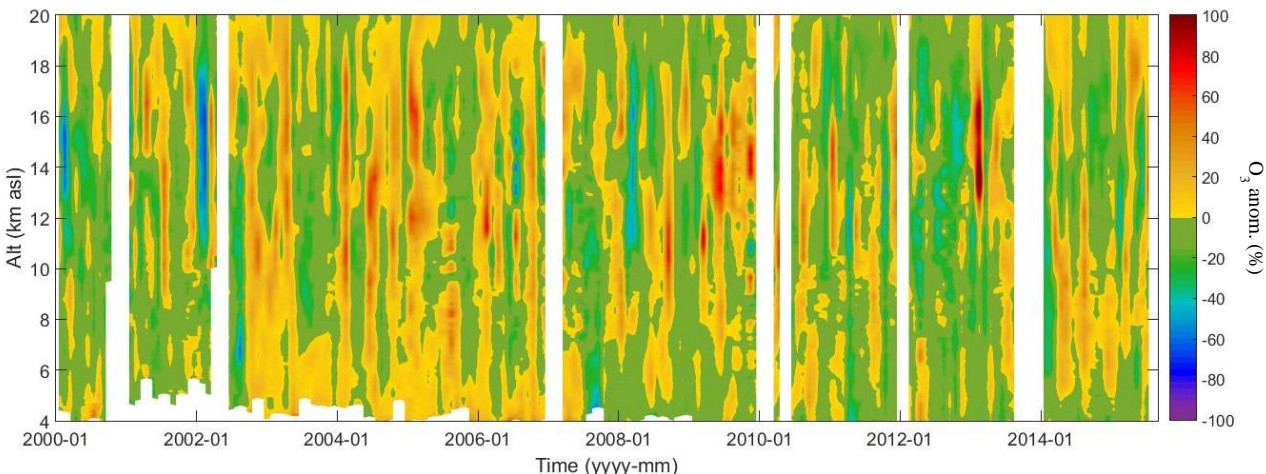

Figure 5. Deseasonalized ozone mixing ratio above TMF. Anomalies (in %) were computed with respect to the climatological (2000-2015) monthly mean





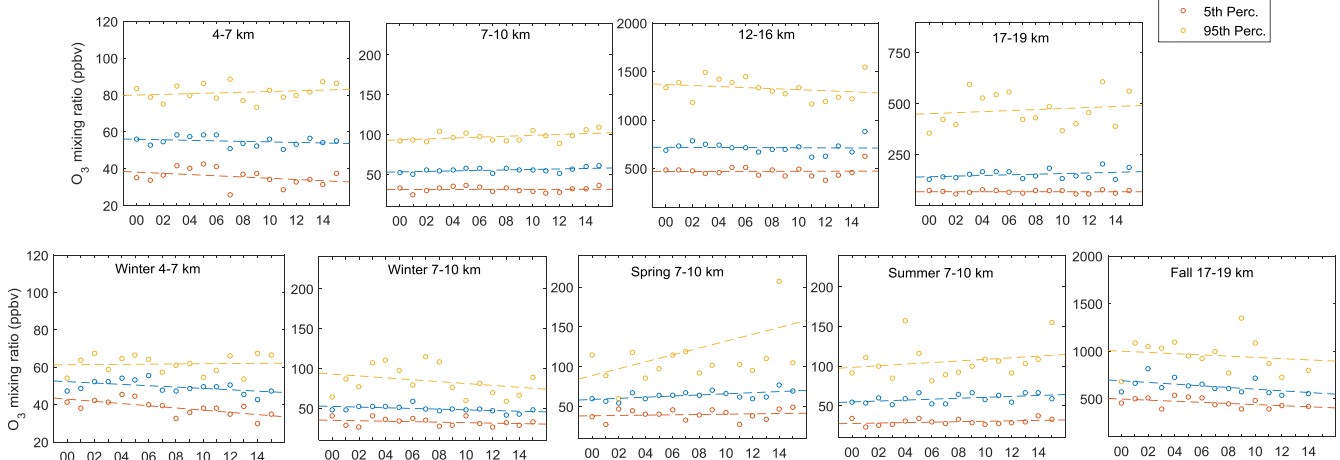

Figure 6. Time series of the median (blue), 5th (orange) and 95th (yellow) percentile ozone values at different altitude layers for the full year (top) and for selected seasons and altitude layers (bottom) obtained from the TMF lidar measurements. Dashed lines represent the linear fit for each time series





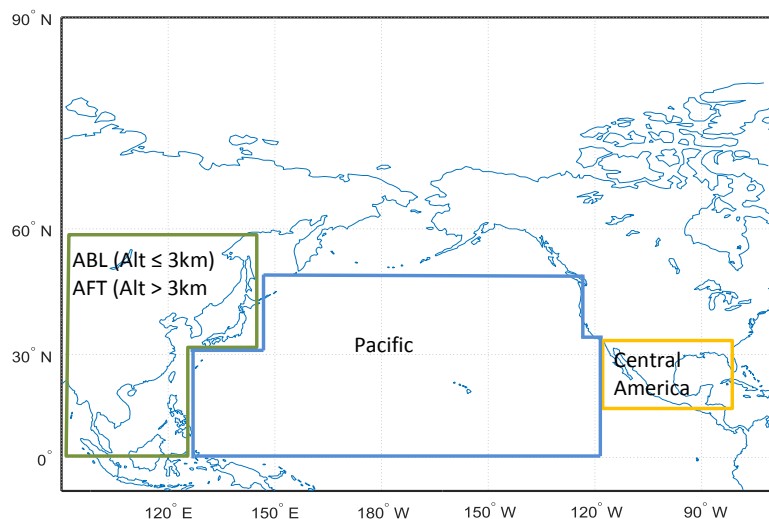

Figure 7. Geographical boundaries used to characterize the air parcels associated with the 8-day backward trajectories ending at TMF during the lidar measurements over the period 2000-2015




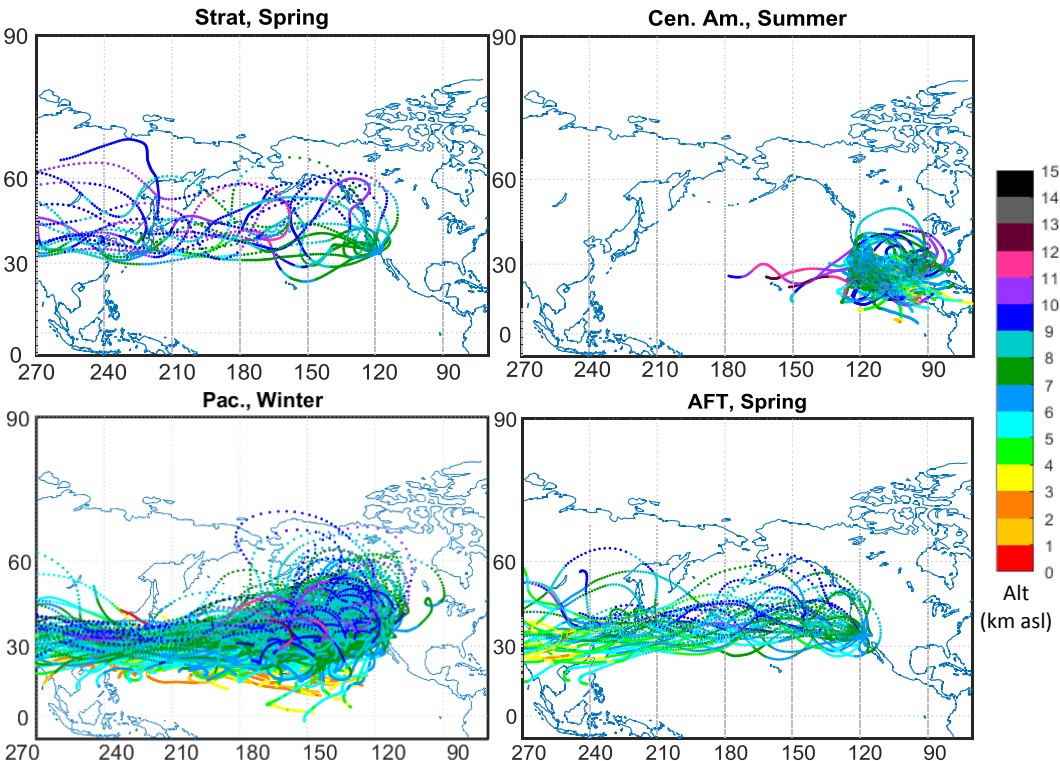

Figure 8. Examples of HYSPLIT 8-day backward trajectories arriving at TMF at 7 km altitude
for four selected seasons and categories (see text for details)




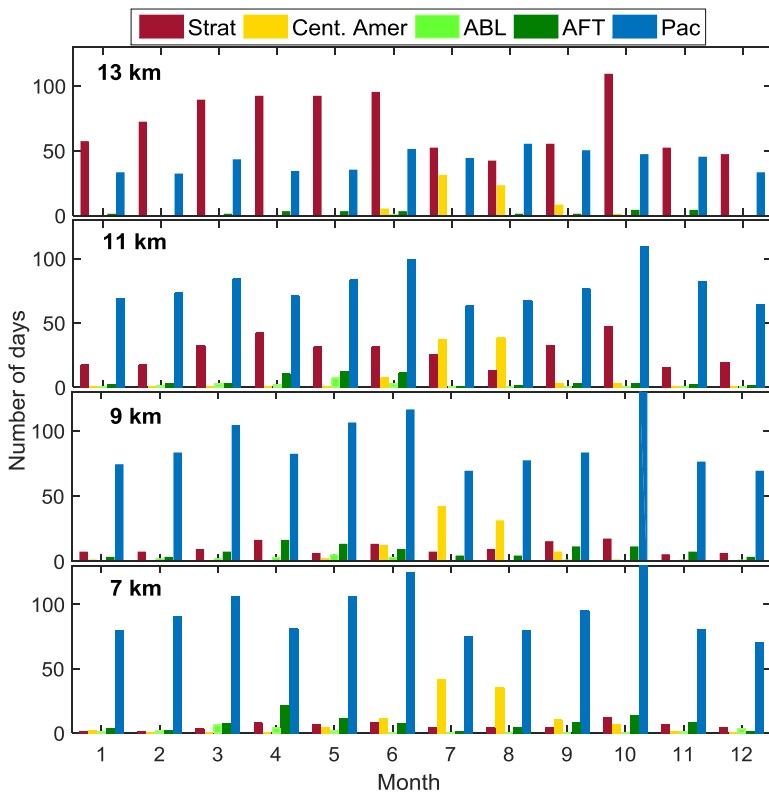

Figure 9. Distribution of the five categories identified for each trajectory ending at TMF during the lidar measurements over the period 2000-2015. The number of occurrences is given for each month of the year, and for four different altitude layers





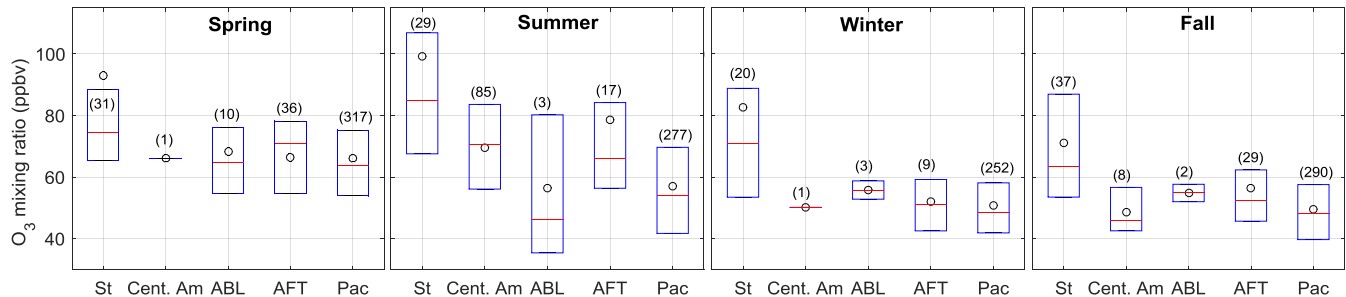

Figure 10. Box plot of the ozone mixing ratios measured within the air masses arriving at TMF at 9 km for the five identified categories (see text for details) and the four seasons. The black dot represents the mean value, the red line is the median and the box limits correspond to the 25[th] and 75[th] percentiles. The numbers between parentheses indicate the number of associated trajectories





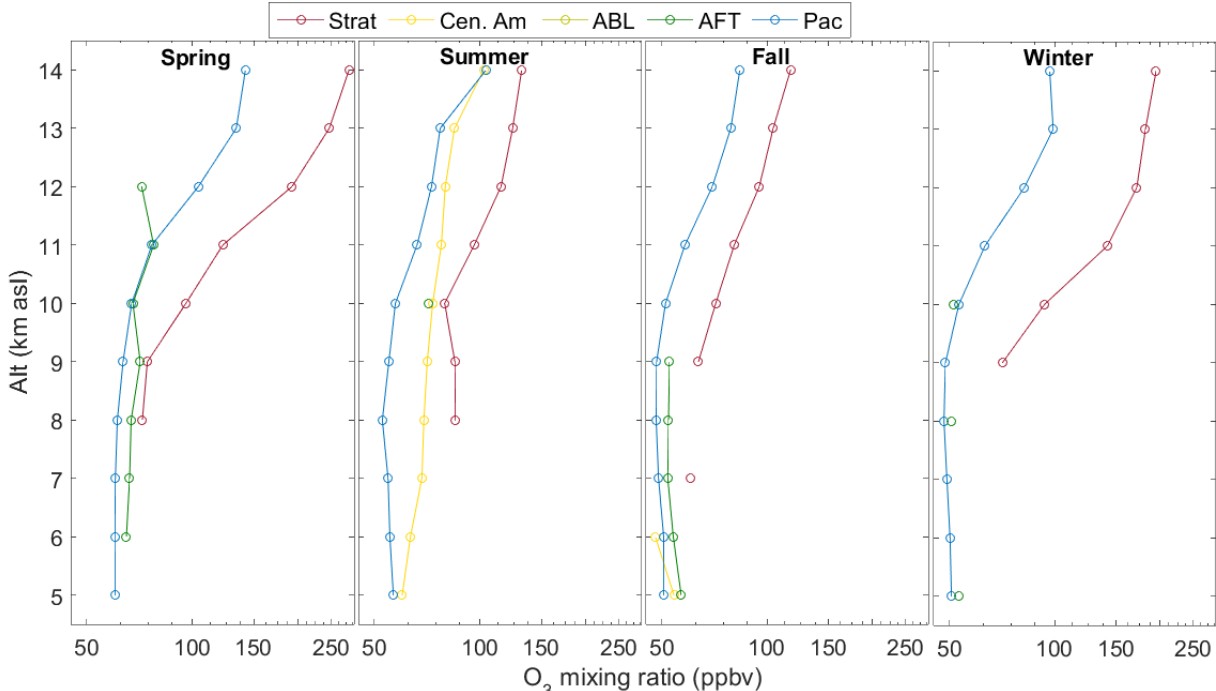

Figure 11. Composite profiles of the ozone mixing ratio associated with the different categories and for each season. Results are shown only when the number of samples for a given category was larger than 5% of the total number of samples in that season



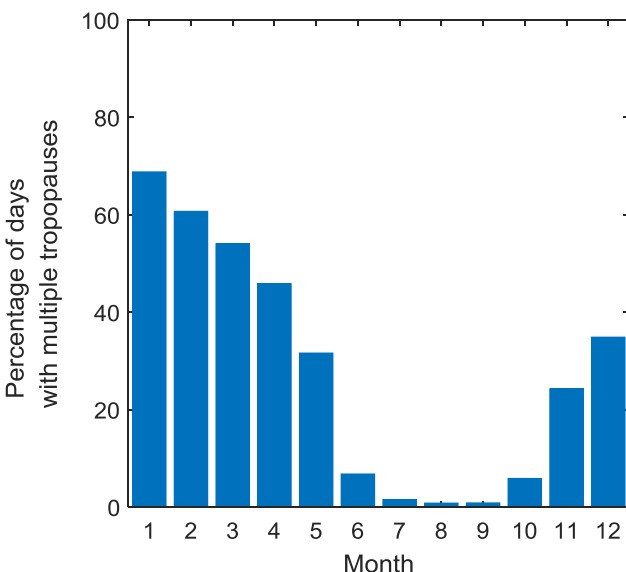

Figure 12. Monthly distribution of occurrences (in %) of double tropopauses above TMF. The number of days with tropopause folds is normalized to the total number of measurements per month compiled in Table1





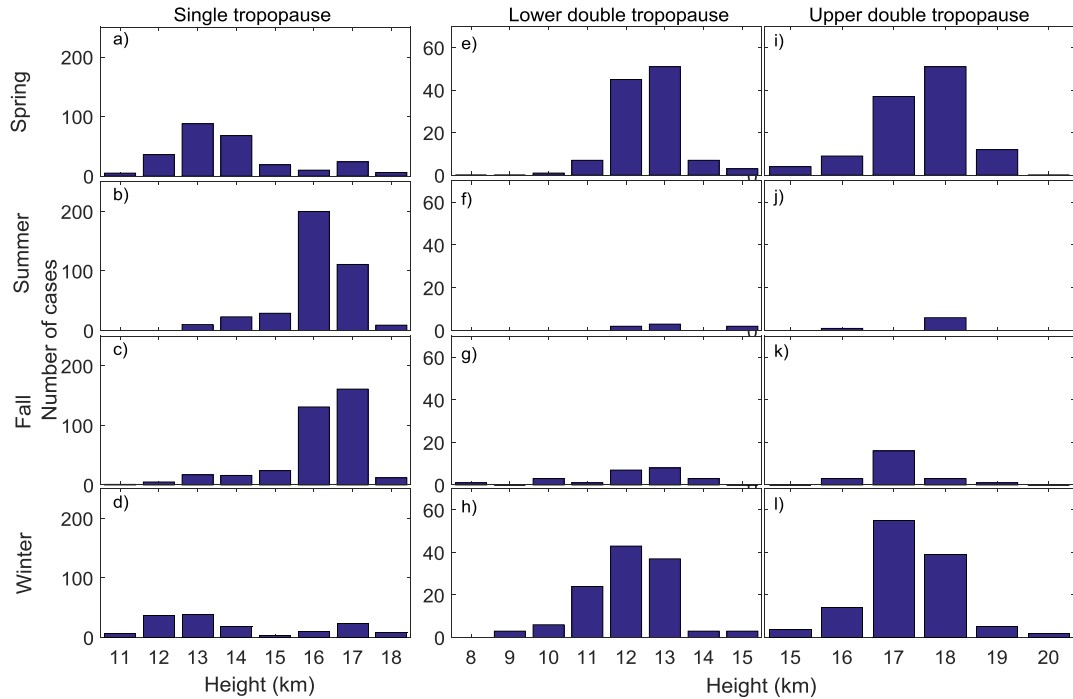

Figure 13. a) to d) Altitude distribution of the tropopause above TMF for spring, summer, fall and winter respectively, and in the absence of double-tropopause. e) to h) Altitude distribution of the lower (first) tropopause above TMF for spring, summer, fall and winter respectively, and in the presence of a double-tropopause. i) to l) Same as e) to h) but for the upper or second tropopause. All computations were made at the times of the TMF lidar measurements





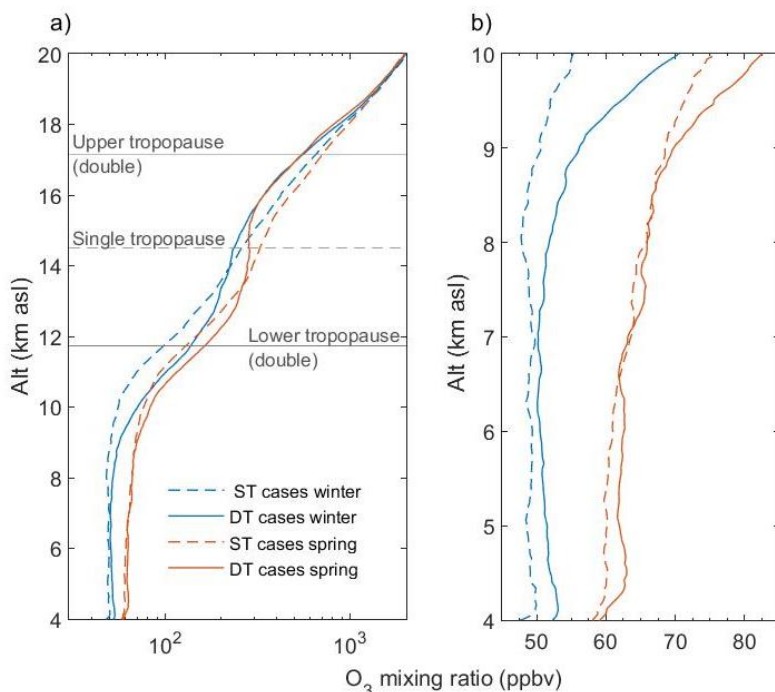

Figure 14. Winter- (cyan) and Spring- (red) averaged ozone mixing ratio profiles computed in the presence of a double tropopause (DT, solid curves) and single tropopause (ST, dashed curves). The horizontal solid grey curves depict the average altitude of the lower and upper tropopauses when a double tropopause was identified. The horizontal dashed grey line corresponds to the average altitude of the tropopause when a single tropopause was identified. b) Same as a) but zoomed on the tropospheric part of the profiles (4-10 km)