# Peer review of "Tropospheric Ozone Seasonal and Long-term Variability as seen by lidar and surface measurements at the JPL-Table Mountain Facility, California"

_Atmospheric Chemistry and Physics, 2016_

## Short Comment (SC1) · 28 Mar 2016

M.-Y. Lin

meiyun.lin@noaa.gov

1. Line 80-85: The discussions on the drivers of tropospheric ozone variability are somewhat incomplete.

Please consider adding a few sentences to describe the role of climate variability and associated changes in atmospheric circulation patterns in contributing to tropospheric ozone interannual variability and decadal trends, as found in the 40-year ozone record at Mauna Loa Observatory in Hawaii (Lin et al., 2014, Nature Gescience).

Meiyun Lin, L.W. Horowitz, S. J. Oltmans, A. M. Fiore, Songmiao Fan (2014): Tropospheric ozone trends at Manna Loa Observatory tied to decadal climate variability, Nature Geoscience, 7, 136-143, doi:10.1038/NGEO2066

2. In the Introduction section, you might also want to add a few overview sentences on the key role of tropopause folds on interannual variability of free tropospheric and surface ozone over the western US (e.g. Lin et al., 2015a, Nature Communications). The literature review will help in placing your Results Sections 3.2 and 3.3. into context.

3. Line 490-493: Regarding the influence of sampling biases on calculated ozone trends, it seems like that you are discussing the results from Lin et al. (2015b, GRL). But the paper is not included in the list of references.

Meiyun Lin, L.W. Horowitz, O.R. Cooper, D. Tarasick, S. Conley, L.T. Iraci, B. Johnson, T. Leblanc, I. Petropavlovskikh, E.L. Yates (2015): Revisiting the evidence of increasing springtime ozone mixing ratios in the free troposphere over western North America, Geophysical Research Letter, 42, doi:10.1002/2015GL065311

4. Line 500-503: Discussions here are somewhat awkward. Are you talking about long-term trends or seasonal variability? I don't believe anyone has suggested that the negative trends (if any) in wintertime ozone over the western US are due to a decrease in background ozone.

5. Line 510: None of the cited references has explicitly discussed the influence from stratospheric intrusions. You should cite other more relevant papers.
* * *

---

## Referee Comment (RC1) · Anonymous Referee #1 · 13 Apr 2016

MS Number : acp-2016-70

Summary :

This paper by Maria Jose Granados-Munoz and Thierry Leblanc presents an interesting analysis of tropospheric ozone profiles over the JPL-Table Mountain Facility, California, based on 16 years of regular Lidar measurements and 2 years of co-located surface measurements. The manuscript spans different aspects of what such an extensive data set allows in terms of characteristics of ozone vertical distribution : diurnal and seasonal cycles, vertical variability from 2 to 20 km, interannual variability and

trends between 2000 and 2015, classification of air mass origins, and influence of tropopause folds.

Authors are right, such extensive ozone data set from JPL-TMF is "very valuable due to the rising interest in the detection of long-term trends in the Western United States . . ." as written in the introduction. Such extensive tropospheric ozone data set has not been presented in publication before and thus deserves a reference paper. The title, as well as the last sentences of the introduction are promising and gives expectations on the understanding of the observed long-term variability. However, the manuscript misses some diagnostics and solid conclusions to provide a thorough assessment to be used by a broad audience. It seems that the manuscript is a compilation of independent sub-sections and the primary objective of understanding the long-term vertical variability has been lost at the end. Subsection 3.5 in particular does not help at all in understanding the interannual variability. Similar remark applies for sub-section 3.4, which is also leading to unexpected conclusion on "No outstanding influence from Asia" as highlighted in the abstract. As a consequence, the discussion and concluding remarks sections do not appear robust enough at this stage of the manuscript. They are sometimes in contradiction with previous findings and let the reader with a promise of a following paper after additional analysis is performed. It's unfortunate.

Regarding the presentation of the manuscript, it is also not very comfortable to have to wait the discussion section to read conclusions and explanation on what have been observed and described in previous sub-sections. I would strongly recommend to include the different paragraphs of the section 4 in their associated sub-sections 3.

Undoubtedly, this manuscript is meant to be a valuable contribution, especially because it is an impressive long-term data set but I have major concerns regarding the robustness of some analysis and conclusions. Therefore, I will recommend publication only after (major and minor) revisions as detailed below are considered. Finally, I support what M.-Y. Lin has posted as an interactive comment and recommend the authors to also follow her recommendations.

Major comments :

-Abstract, line 53: This last sentence "No outstanding influence from Asia was identified" is quite surprising, in contradiction with previous studies, and raises issues on the robustness of the methodology. See further comments below.

- Line 99 : The characteristic "only surface or column-integrated measurements" does not apply to aircraft (MOZAIC/IAGOS) platforms. Zbinden et al., (2013) give a recent detailed description of ozone vertical profiles by MOZAIC/IAGOS over mostly documented airports by the program since 1995 (including Los Angeles airport). Logan et al., (2012) also used vertical profiles recorded by MOZAIC aircraft in addition to ozone sondes for deriving a global picture of ozone trends over Europe.

- Lines 167 to 177 : This paragraph as well as associated Figure 1 are questionable. It seems surprising to see on figure 1a that the difference is not varying with altitude. Given the argument from the last sentence of the paragraph that sampled air masses are not the same, one could expect a higher difference in regions with higher ozone variability (UTLS compared to free troposphere). It is not the case. Concluding the section with differences within +/- 15% raises the question on the regional representativeness of the JPL-TMF station. This paragraph deserves much more attention and details to give proof on the consistency between those two independent data sets (if that was the objective of such paragraph ?).

- Line 184 : This paragraph omits to give appropriate accuracy/error procedures of this analyzer and reference (at least from previous use of such type of instrument). Information on operating procedures (maintenance, calibration), or at least a reference paper would be appreciated.

- Line 241 : Figure 3a is not restricted to the tropospheric part of the ozone profile. Either change this adjective in the sentence or the figure. The title of the paper and of the sub-section is clearly on tropospheric ozone variability. I suggest to plot Figure 3 only for altitudes up to 15 km (on average) and modify the X axis for linear concentrations

on the tropospheric typical range [0-150 ppb]. In the seasonal part of the Figure 3, that will give the advantage to further see differences between the averaged seasonal vertical profiles. Is the spring profile that close to the summer profile ? From Figure 4, it seems that the spring profile should highlight higher ozone that the summer one.

- Lines 268-270 : The spring-summer maximum is indeed a common characteristic of northern mid-latitudes over Europe and North-America. However, it seems that these JPL-TMF data as shown in Figure 4 shows a clear spring maximum (at least April-May). This seems actually consistent with what Zbinden et al, (2013) have shown over Los Angeles airport based on MOZAIC data. This paragraph deserves further clarification to highlight this "local" characteristic of a clear spring maximum (if confirmed by Figure 3 plotted with a tropospheric X axis only). This special feature is likely the result of stronger influence from Asia in spring over the west coast of the US (Jaffe et al.2003; Parrish et al., 2004; Cooper et al., 2005; Neuman et al., 2012) as mentioned by Zbinden et al., (2013). Therefore, it is very surprising to read in the abstract that "No outstanding influence from Asia was identified". I am not convinced by such statement. Further comments below regarding subsection 3.4.

- Lines 302-311 : As far as I know, the common rule for indicator of the statistical significance of the trends with such procedure is to use p-Values lower than 0.05 for trends with a confidence level larger than 95% (and not p larger than 0.1 and confidence level larger than 90%). I recommend the authors to rewrite tables 2 and 3 applying these criteria. It turns out that only spring (in the upper troposphere) and winter (from 4 to 10 km) show significant seasonal trends. It would also be nice to give indication of error in % in first part of Table 3 as done in Table 2.

- Lines 322-328 : This decrease in winter is the most surprising information of this paper. As far as I know, there is no other reference mentioning such significant decrease in winter in the region. In the recent review paper of Cooper et al., 2014 (and references therein including Cooper et al., (2012)), reported stations in California do show significant increase in winter. Therefore I recommend the authors to further argue

and explain why such a different behavior at JPL-TMF station. This entire subsection misses further discussion on this trends analysis. See additional comments below regarding section 4.

- Sub-section 3.4 : This section is the weakest point of the draft paper. My major concern can be summarized as 2 questions : why (only) 8-days backward trajectories ? why (so many) 2-days of residence time over Asia ? These arbitrary choices need further arguments, sensitivity analysis and/or references for similar studies. From my knowledge, Cooper et al, (2010) used 15-days backward trajectories and concluded with a significant influence from Asia to the Western US region, especially in spring. To me, the length of the trajectories used here may be too short and the classification criteria is not adapted, to really assess the different influences authors are looking for. The two plots on Figure 8, bottom row, are too similar to trust in this methodology of classification. The criteria for air masses to be classified as ABL or AFT (at least 2 days over there) is too difficult to meet. What is the reason to impose 2 days over the continent to classify air parcels as "Asian" ? It is way too much. This is probably the reason why the Asian influence does not appear as strong as expected from previous studies. I guess that some (or most of) the air masses classified as Pacific (especially in Spring) would be classified Asian with a different criteria. Results and conclusions may be different with longer trajectories and different criteria for classification. For example, I would suggest to change the order of the sequential attributions : By default, air parcels that are not "stratosphere" would be classified as "Asian" unless the trajectories spend the entire period (8 days or longer) over the Ocean or over Central America. Maybe I'm wrong but I recommend the authors to revise this sub-section to make it convincing. Besides, an interesting information from this analysis of classified air masses would be to check if there is a tendency or anomalous behaviors from one year to another. If I understand well the title and the introduction, this subsection should help further understand the trend analysis. This is not the case so far.

- Subsection 3.5 : A figure showing an individual profiles (not averaged as in Fig.14)

with double tropopause would be good to further explain this characteristic to the non-expert reader. This section should make the distinction clearer between the wording "double tropopause" and "tropopause folds". It is not the same. Sentence line442-443 and legend of Figure 12 are ambiguous. Indeed, Randel et al., (2007) should be cited as a reference paper for characteristics of double tropopauses.

- Section 4 : This section is a bit too long. It starts like a summary but then includes analysis and explanations that would be better placed before in the associated sub-sections. More importantly, some highlighted discussions are mentioned as in contradiction with other studies without any further discussions. This needs to be further argued. For example :

- Line 477-484: Either the comparison with recent findings by Lin et al., (2015) and Neu et al., (2014) makes senses and the inconsistency raises important questions (i.e., what makes JPL-TMF station different and not representative of the general behavior ?), or the comparison is not possible (need to explain why ?) and such paragraph is simply to be removed.

- Lines 498-503 : This negative trend in winter observed at JPL-TMF is surprising, in contradiction with most of the studies I'm aware of, and therefore deserves further investigations. According to Cooper et al., (2012) only one site in the Western US shows negative trend in winter, and only for the 95% percentile. This is different from what is presented in Table 2 in this study. "Decrease in background ozone during these months . . . " is definitely an opposite conclusion to that of Cooper et al., (2012) and of similar recent studies. As for previous comment, I recommend the authors to answer the question "why do JPL-TMF measurements highlight different behaviors ?"

- Line 513-517 : Wouldn't it be good to have results from this extended analysis ?

Minor comments :

- Line 61 : "directly emitted" seems to me too ambiguous and may let the reader think

that ozone is a primary pollutant. I suggest to replace by "transported down from the stratosphere".

- Line 88-89 : I suggest to remove the end of the sentence starting at "which has not yet been . . ." because this not true, as written indeed in the following paragraph.

- Line 285-286 : Do R=0.34 or 0.44 really show correlation ?

- Line 287-290 : This is more interesting than the lines before and deserves a brief explanation. Why outliers have to be removed to confirm the correlation. Does this tell us something on specific process at the surface or at 4-6 km altitude? Is there different disconnected influences ? For a specific season ?

- Figure 5 : I am wondering if the choice of the color sale is the most appropriate to highlight (real and significantly positive or negative) anomalies. What about choosing a color scale centered on 0 (same color for -10 to +10) ? It is difficult to check consistency with Figure 6. Is there any explanation for long-lasting anomalies in 2003-2007 ?

- Line 341 : This reference is missing in the list.

- Line 412-414 : This sentence needs to be accompanied by a reference to give argument that an excess of 15 ppbv is what is expected as lightning-induced enhancement of ozone. Is Cooper et al. (2009) as mentioned in section 4 (line 537) relevant for this ?

- Line 429 : MERRA acronym needs to be explained. Reference would be nice.

- Reference list : The following references are incomplete : Ambrose et al., 2011; Cooper and Stohl, 2005; Lee and Akimoto, 1998; Monks, 2005; Petetin et al., 2015 (should check if ACP reference is available).

References :

Cooper, O. R., Stohl, A., Eckhardt, S., Parrish, D. D., Oltmans, S. J., Johnson, B. J., Nédélec P., Schmidlin, F. J., Newchurch, M. J., Kondo, Y., and Kita, K.: A springtime

comparison of tropospheric ozone and transport pathways on the east and west coasts of the United States, J. Geophys. Res., 110, D05S90, doi: 10.1029/2004JD005183, 2005.

Jaffe, D., Price, H., Parrish, D., Goldstein, A., and Harris, J.: Increasing background ozone during spring on the west coast of North America, Geophys. Res. Lett., 30, 1613, doi:10.1029/2003GL017024, 2003.

Logan J.A., J. Staehelin, I. A. Megretskaia, J.-P. Cammas, V. Thouret, H. Claude, H. De Backer, M. Steinbacher, H. E. Scheel, R. Stübi, M. Fröhlich, and R. Derwent, Changes in ozone over Europe since 1990: analysis of ozone measurements from sondes, regular aircraft (MOZAIC) and alpine surface sites. J. Geophys. Res., D09301, doi:10.1029/2011JD016952, 2012.

Neuman, J. A., Trainer, M., Aikin, K. C., Angevine, W. M., Brioude, J., Brown, S. S., de Gouw, J. A., Dube, W. P., Flynn, J. H., Graus, M., Holloway, J. S., Lefer, B. L., Nedelec, P., Nowak, J. B., Parrish, D. D., Pollack, I. B., Roberts, J. M., Ryerson, T. B., Smit, H., Thouret, V., and Wagner, N. L.: Observations of ozone transport from the free troposphere to the Los Angeles basin, J. Geophys. Res., 117, D00V09, doi: 10.1029/2011JD016919, 2012.

Parrish, D., Dunlea, E. J., Atlas, E. L., Schauffler, S., Donnelly, S., Stroud, V., Goldstein, A. H., Millet, D. B., McKay, M., Jaffe, D. A., Price, H. U., Hess, P. G., Flocke, F., and Roberts, J. M.: Changes in the photochemical environment of the temperate North Pacific troposphere in response to increased Asian emissions, J. Geophys. Res., 109, D23S18, doi: 10.1029/2004JD004978, 2004.

Randel, W. J., D. J. Seidel, and L. L. Pan, Observational characteristics of double tropopauses, J. Geophys. Res., 112, D07309, doi:10.1029/2006JD007904, 2007.

Zbinden R.M., V. Thouret, P. Ricaud, F. Carminati, J.-P. Cammas, and P. Nédélec, Climatology of pure tropospheric profiles and column contents of ozone and carbon

monoxide using MOZAIC in the mid-northern latitudes (24° N to 50° N) from 1994 to 2009„ Atmos. Chem. Phys., 13, 12363-12388, doi:10.5194/acp-13-12363-2013, 2013.

---

## Referee Comment (RC2) · M. J. Newchurch (Referee) · 2 May 2016

General Comments: This paper is a very useful contribution to the rather limited literature of the vertical distribution of ozone over the USA west coast. The analytical techniques employing trajectory analysis, stratospheric-tropospheric folding dynamical structures, time series and variability analysis, and attribution analysis all contribute to the value of this work. The major shortcomings concern the choices for trajectory/attribution parameters (primarily time scales) and the absence of a concise conclusion section. Minor shortcomings concern the details of the trend analysis and some

none inconsistencies in the attribution. This paper should be accepted after these issues are successfully addressed.

Specific Comments: L36: 'No outstanding influence from Asia was identified'. This absence of Asian influence is strongly dependent on the somewhat arbitrary selection of trajectory time-scale parameters. This conclusion is also somewhat inconsistent with the early spring maximum in figure 4. Consider additional analyses to resolve this discrepancy by providing compelling evidence to support your finding. L44: 'Tropospheric ozone can be directly emitted to the troposphere, ': Direct emissions (separate from STE injections) are a very small fraction of tropospheric ozone sources. Suggest you omit this sentence. L273: Removing data +/- 1 sd for a correlation calculation is not a legitimate approach. That process will remove approximately 1/3 of the data and will certainly enhance the correlation between the remaining data, but one cannot justify removing that many data and one would certainly not call all those data 'outliers'. L296: Suggest you use p-values of 0.05 to be consistent with the 95% statistics used elsewhere. Section 4: The summary should be expressed in the Abstract. No need for another summary here. The more discussions should be moved to the section under discussion or a new section heading inserted. The paper needs a short 'Conclusions' section (not summary or discussion). The conclusions should be succinct and describe the main take-home points derived from the paper.

Technical corrections: See attached .docx for suggested tracked changes.

Please also note the supplement to this comment:
http://www.atmos-chem-phys-discuss.net/acp-2016-70/acp-2016-70-RC2-supplement.pdf

[Figure]

**Supplement:**

[revised manuscript text omitted]

4.  Summary and discussion

Long-term (2000-2015) tropospheric ozone measurements obtained from the differential
absorption lidar located at the JPL Table Mountain Facility in Southern California (elev. 2285 m)
were used for the first time to investigate ozone variability at time-scales ranging from days to
decades. The routine nighttime lidar profiles (3.5-25 km) were complemented by 24-hour
continuous surface ozone measurements acquired since 2013. The present study allowed the
characterization of the full profile from the ground to the stratosphere, and is  particularly
valuable in the context of a notoriously sparse horizontal coverage for this type of vertically
resolved measurements in a region affected by transboundary ozone inflow and stratospheric
intrusions.

The observed diurnal and seasonal variability is consistent with that observed at other high-
elevation rural sites, implying high values and low amplitudes of the seasonal and diurnal cycles
compared to those observed at low elevation and/or urban sites (Brodin et al., 2010; Gallardo et
al., 2000; Naja et al., 2003). The monthly mean surface ozone mixing ratio agrees well with that
measured by lidar in the lower troposphere throughout the year (Figure 4), thus revealing that
nighttime surface conditions are representative of nighttime lower tropospheric conditions.
Additional daytime lidar measurements will be performed in 2016 to assess whether such
consistency also exists at other times of the day, especially in the afternoon.

The analysis of the long-term lidar time-series (16 years covered) shows no significant
signatures of interannual variability, though larger ozone values were generally observed in
2003-2007 compared to the rest of the measurement period. More importantly, no obvious
signature of ENSO or the QBO could be identified, which is inconsistent with the recent findings
of Lin et al., (2015) or Neu et al., (2014), who have linked tropospheric ozone variability in the
Northern Hemisphere to El Niño/ La Niña events, and the QBO, through the variations of
stratospheric/tropospheric ozone fluxes. However, this inconsistency might not be so surprising
if we take into account the obvious difference in measurement sampling (one single location in
the Western U.S. as opposed to global observations). Nevertheless, our analysis reveals
statistically-significant trends for selected seasons and altitudes. Specifically, a positive trend of
0.31 ppbv.year$^{-1}$ (ozone annual median) was found in the upper troposphere (7-10 km). This
positive trend is more pronounced in spring and summer (0.71 and 0.58 ppbv.year$^{-1}$
respectively), whereas a negative trend (-0.43 ppbv.year$^{-1}$) was found in winter. A negative trend
of -0.36 ppbv.year$^{-1}$ was also detected in winter in the lower troposphere (4-7 km). <These
results should include trend uncertainties and should include only significant figures
in the trend estimates. The wintertime negative results require an explanation.> Our
springtime positive trend estimates only broadly agree with those reported in the literature. The
two-decade-long efforts to implement regulations to control air quality and anthropogenic
emissions in the U.S. have led to a clear decrease in ozone levels in the Eastern U.S., but not in
the Western U.S. (e.g. Copper et al., 2012; 2014). This different regional behavior has been
attributed to the inflow of elevated ozone, mainly from East Asia, and to the increasing
contribution of stratospheric intrusions (Cooper et al., 2010; Jacob et al., 1999; Parrish et al.,
2009; Reidmiller et al., 2009). But again, differences in sampling can impact significantly the
interpretation of our trend estimates. As pointed out by Lin et al. (2015b), further coordination
efforts at both global and regional scales are necessary in order to reduce biases introduced by
inhomogeneity in sampling. As part of these efforts, an extended analysis based on the origin of
the air masses sampled by the TMF lidar is under way, with the ultimate objective to filter
out synoptic noise out, and better quantify the ENSO and QBO signals and the residual trends.
       As a prerequisite to such study, a preliminary characterization of the air masses sounded by the
       TMF  lidar was performed and presented here.

Backward trajectories were computed in an attempt to classify the air masses sampled by the
TMF lidar over the period 2000-2015. Our classification, based on a 2-day <Perhaps too long a
requirement?> residence time within each   region,   comprised   five   categories,   namely
       "stratosphere", 'Pacific Ocean", "Asian

Boundary Layer", "Asian free-troposphere", and "Central America". After computing more than
20,000 8-day back-trajectories covering the troposphere every 1–km for each lidar measurement
made between 2000 and 2015, the Pacific Ocean and the stratosphere categories were found
most frequently. Not surprisingly, parcels identified as "stratosphere" contain the highest ozone
mixing ratios, and parcels classified as "Pacific Ocean" contain the lowest ozone concentrations,
which can be considered as 'background conditions'. No outstanding signature from the Asian
boundary layer could be identified due to the low number occurrences associated with this
category. Nevertheless, air parcels classified as "Asian free-troposphere" seemed to contain
systematically more ozone than those classified as "Pacific Ocean", especially below 9 km. A
refined classification, probably requiring the use of chemistry-transport models, is needed to
assess whether the Asian boundary layer or the Asian free-troposphere have a detectable impact
on the ozone content measured above TMF.

In summer, air masses coming from Central America were frequently detected. The ozone
mixing ratio values measured in this case were clearly above the climatological mean, with
values up to 15 ppbv larger than those associated with the Pacific Ocean region. Previous studies
(Cooper et al., 2009), have observed enhanced ozone values associated with the North American
Monsoon, mainly due to ozone production associated with lightning (Choi et al., 2009; Cooper et
al., 2009). However, this feature was observed in the Eastern U.S. Because of the synoptic
conditions during the monsoon, the Western U.S. is not as much influenced and no significant
ozone increase was reported (Barth et al., 2012; Cooper et al., 2009). Nevertheless, Cooper et al.,
(2009) reported higher modeled lightning-induced $NO_x$ concentrations at TMF than at other
western locations, which would be consistent with our findings. Further investigation, including
a detailed history of the meteorological conditions along the trajectories, is needed to confirm
this correlation.

As part of the characterization of the TMF lidar-sampled air masses, the impact of double
tropopauses was assessed. The presence of a double tropopause was identified using the MERRA
temperature profiles interpolated at the location of TMF. Between 2000 and 2015, the frequency
of occurrence of double tropopauses above TMF was found to be around 27%. This high
frequency, especially observed in winter and spring, was expected considering the latitude of
TMF, i.e., near the subtropical jet, where frequent tropopause folds occur. More interestingly, a clear, dual vertical structure in ozone was observed in the presence of a double tropopause
(Figure 14). The first (lower) tropopause, which typically coincides with the WMO tropopause
reported by meteorological services, was systematically identified around 12-14 km. The second

[revised manuscript text omitted]

---

## Author Comment (AC1) · 13 Jun 2016

M. J. Granados-Muñoz and T. Leblanc

mamunoz@jpl.nasa.gov

We would like to thank Dr Meiyun Lin for her comments and suggestions. Please, find below the detailed responses.

Comments: 1. Line 80-85: The discussions on the drivers of tropospheric ozone variability are somewhat incomplete. Please consider adding a few sentences to describe the role of climate variability and associated changes in atmospheric circulation patterns in contributing to tropospheric ozone interannual variability and decadal trends, as found in the 40-year ozone record at Mauna Loa Observatory in Hawaii (Lin et al.,

2014, Nature Gescience).

Meiyun Lin, L.W. Horowitz, S. J. Oltmans, A. M. Fiore, Songmiao Fan (2014): Tropospheric ozone trends at Manna Loa Observatory tied to decadal climate variability,Nature Geoscience, 7, 136-143, doi:10.1038/NGEO2066

2. In the Introduction section, you might also want to add a few overview sentences on the key role of tropopause folds on interannual variability of free tropospheric and surface ozone over the western US (e.g. Lin et al., 2015a, Nature Communications) The literature review will help in placing your Results Sections 3.2 and 3.3. into context.

Response:

The following sentences have been added to the introduction:

"Additional factors that have been observed to influence tropospheric ozone variability are climate variability and related global circulation patterns such as ENSO or PDO (e.g. Lin et al., 2014; 2015a; Neu et al., 2014). Tropopause folds also play a key role on tropospheric ozone interannual variability, as they influence the ozone budget in the troposphere and can even affect air quality near the surface (e.g. Lin et al., 2015a Brown-Steiner and Hess, 2011; Langford et al., 2012)."

Comments:

3. Line 490-493: Regarding the influence of sampling biases on calculated ozone trends, it seems like that you are discussing the results from Lin et al. (2015b, GRL). But the paper is not included in the list of references.

Meiyun Lin, L.W. Horowitz, O.R. Cooper, D. Tarasick, S. Conley, L.T. Iraci, B. Johnson, T. Leblanc, I. Petropavlovskikh, E.L. Yates (2015): Revisiting the evidence of increasing springtime ozone mixing ratios in the free troposphere over western North America, Geophysical Research Letter, 42, doi:10.1002/2015GL065311

Response:

The paper was missing from the reference list. It is now included.

Comment: 4. Line 500-503: Discussions here are somewhat awkward. Are you talking about long-term trends or seasonal variability? I don't believe anyone has suggested that the negative trends (if any) in wintertime ozone over the western US are due to a decrease in background ozone.

Response: Discussion refers to long term trends. Negative trends were observed at some stations in Cooper et al., (2012) during wintertime, even though most of them were not significant. A significant negative trend is also observed for the median values in this study from 4 up to 10 km during wintertime. The comment on the background ozone decrease has been removed considering all the comments in this respect.

Comment: 5. Line 510: None of the cited references has explicitly discussed the influence from stratospheric intrusions. You should cite other more relevant papers.

Response: The studies by Cooper et al., 2010 and Parrish et al., 2009, provide information about the different regional behavior observed between the western and the eastern US regarding ozone trends. These studies explain the causes for this difference, including both the Asian transport and stratospheric influence affecting predominantly the Western region. References Lin et al., (2012a; 2015a) and Lefohn et al., (2011; 2012) have been included for completeness.
* * *

---

## Author Comment (AC3) · 13 Jun 2016

MS Number: acp-2016-70

Summary:

This paper by Maria Jose Granados-Munoz and Thierry Leblanc presents an interesting analysis of tropospheric ozone profiles over the JPL-Table Mountain Facility, California, based on 16 years of regular Lidar measurements and 2 years of co-located surface measurements. The manuscript spans different aspects of what such an extensive data set allows in terms of characteristics of ozone vertical distribution: diurnal and seasonal cycles, vertical variability from 2 to 20 km, interannual variability and trends between 2000 and 2015, classification of air mass origins, and influence of tropopause folds.

Authors are right, such extensive ozone data set from JPL-TMF is "very valuable due to the rising interest in the detection of long-term trends in the Western United States : : :" as written in the introduction. Such extensive tropospheric ozone data set has not been presented in publication before and thus deserves a reference paper. The title, as well as the last sentences of the introduction are promising and gives expectations on the understanding of the observed long-term variability. However, the manuscript misses some diagnostics and solid conclusions to provide a thorough assessment to be used by a broad audience. It seems that the manuscript is a compilation of independent sub-sections and the primary objective of understanding the long-term vertical variability has been lost at the end. Subsection 3.5 in particular does not help at all in understanding the interannual variability. Similar remark applies for sub-section 3.4, which is also leading to unexpected conclusion on "No outstanding influence from Asia" as highlighted in the abstract. As a consequence, the discussion and concluding remarks sections do not appear robust enough at this stage of the manuscript. They are sometimes in contradiction with previous findings and let the reader with a promise of a following paper after additional analysis is performed. It's unfortunate.

Regarding the presentation of the manuscript, it is also not very comfortable to have to wait the discussion section to read conclusions and explanation on what have been observed and described in previous sub-sections. I would strongly recommend to include the different paragraphs of the section 4 in their associated sub-sections 3. Undoubtedly, this manuscript is meant to be a valuable contribution, especially because it is an impressive long-term data set but I have major concerns regarding the robustness of some analysis

**and conclusions. Therefore, I will recommend publication only after (major and minor) revisions as detailed below are considered. Finally, I support what M.-Y. Lin has posted as an interactive comment and recommend the authors to also follow her recommendations.**

Response:

First of all, we would like to thank the reviewer for the comments and suggestions which have helped to improve the quality of the manuscript. Especially, section 3.4 is now much improved and more significant results were found. Comments and suggestions have been really helpful to improve the section and the quality of the manuscript.

The study presents the tropospheric ozone lidar database acquired at TMF, which has not been previously presented in a peer reviewed publication and we consider it is of high interest due to its characteristics. As the reviewer says, the data set allows characterizing the diurnal and seasonal cycles, vertical variability, interannual variability and trends and the influence of different sources affecting the ozone profiles measured at the station. Sections 3.4 and 3.5 are mainly intended to contribute to the characterization of tropospheric ozone above TMF and to analyze the influence of the different sources on the ozone profiles. However, the goal of using them to understand the long-term variability is too ambitious for this study. Trends related to the different air masses categories presented in Section 3.4 and trends in the tropopause folds and their possible influence on the interannual variability has been investigated, but no statistically significant results were observed because the number of data is drastically reduced when subdivided in smaller datasets. Nonetheless, we think this analysis deserves an additional publication where the lidar database is combined with additional data (e.g. model, satellite, additional stations…) to reach significant results and conclusions in this respect.

Regarding the influence from Asia, section 3.4 has been rewritten based on the new criteria used for the classification of the air masses (see following responses).

Discussion section has now been suppressed and the comments are included in the corresponding subsections of the results as suggested by the reviewer. Recommendations and comments by Meiyun Lin have also been addressed.

**Major comments:**
**-Abstract, line 53: This last sentence "No outstanding influence from Asia was identified" is quite surprising, in contradiction with previous studies, and raises issues on the robustness of the methodology. See further comments below.**

Response:
The abstract has been modified according to the new results obtained in Section 3.4. The sentence is no longer included. Instead, the following sentence has been added:
Lines 55-56: *"Influence from Asia was observed throughout the year, with more frequent episodes during spring, associated to ozone values from 53 to 63 ppbv at 9 km."*

**- Line 99: The characteristic "only surface or column-integrated measurements" does not apply to aircraft (MOZAIC/IAGOS) platforms. Zbinden et al., (2013) give a recent detailed description of ozone vertical profiles by MOZAIC/IAGOS over mostly documented airports by the program since 1995 (including Los Angeles airport). Logan et al., (2012) also used vertical profiles recorded by MOZAIC aircraft in addition to ozone sondes for deriving a global picture of ozone trends over Europe.**

Response:
Information regarding aircraft platforms and the suggested references have been included in the introduction.
Lines 114-117: *"Ozone vertical profiles have also been obtained from aircraft platforms by means of programs such as MOZAIC and IAGOS, available since 1995 (e.g. Zbinden et al., 2013, Logan et al., 2012). However, aircraft data are limited to air traffic routes and the temporal resolution depends on the frequency of the commercial flights."*

**Comment:**
**- Lines 167 to 177: This paragraph as well as associated Figure 1 are questionable. It seems surprising to see on figure 1a that the difference is not varying with altitude. Given the argument from the last sentence of the paragraph that sampled air masses are not the same, one could expect a higher difference in regions with higher ozone variability (UTLS compared to free troposphere). It is not the case. Concluding the section with differences within +/- 15% raises the question on the regional representativeness of the JPL-TMF station. This paragraph deserves much more attention and details to give proof on the consistency between those two independent data sets (if that was the objective of such paragraph ?).**

Response:

The objective of the paragraph is to show the accuracy of the lidar measurements using for validation an independent and reliable technique such as the ECC, since lidar measurements from the DIAL at JPL-TMF have not been presented before. The paragraph has been modified and additional details are now provided.
The effect of the drift of the ozonesonde is not visible in figure 1 due to the larger ozone values observed in the UTLS region compared to the troposphere. We see an increase in the difference when we represent the RMSE (see figure below), but the relative differences expressed in percentage remain almost constant with altitude.
The average difference between the lidar data and the ECC (7% in the troposphere) is within the ranges observed in previous studies (see e.g. Sullivan et al., 2015) for lidar systems and it is a reasonable value considering the differences between both techniques. The 15 % value has been replaced by 10%, which is more accurate in view of the plots in figure 1 and the paragraph has been rewritten.

Lines 180-194: *"The TMF ozone lidar measurements have been regularly validated using simultaneous and co-located Electrochemical Concentration Cell (ECC) sonde measurements (Komhyr, 1969; Smit et al., 2007). In the troposphere the precision of the ozonesonde measurement is approximately 3-5% with accuracy of 5-10% below 30 km. TMF has ozonesonde*

*launch capability since 2005 and 32 coincident profiles were obtained over the period 2005-2013. Results from the lidar and the ECC comparison are included in Figure 1.Figure1a shows the averaged relative difference between the lidar and ECC ozone number density profiles for the 32 cases. On average, the lidar and sonde measurements are found to be in good agreement, with an average difference of 7% in the bulk of the troposphere and most of the values under 10% (Figure 1b), which is within the combined uncertainty computed from both the lidar and sonde measurements. Note that a non-negligible fraction of the differences is due to the different measurement geometry of the lidar and ozonesonde: 2-hour averaged, single location for lidar, and horizontally-drifting 1-second measurements for the ozonesonde usually rising at 5 m·s⁻¹. Figure 1c reveals that the deviations do not present significant changes with time, which is an indicator of the system stability despite the multiple upgrades made over this time period."*

Reference:
Sullivan, John T., Thomas J. McGee, Russell DeYoung, Laurence W. Twigg, Grant K. Sumnicht, Denis Pliutau, Travis Knepp, and William Carrion. "Results from the NASA GSFC and LaRC ozone lidar intercomparison: New mobile tools for atmospheric research." Journal of Atmospheric and Oceanic Technology 32, no. 10, 1779-1795, 2015.

[Figure]

Figure. a) Profile of RMSE for the lidar compared to the ECC ozone number density for the 32 simultaneous measurements (dark blue). The black solid curve shows the number of data points at each altitude.

**Comment:**
**- Line 184: This paragraph omits to give appropriate accuracy/error procedures of this analyzer and reference (at least from previous use of such type of instrument). Information on operating procedures (maintenance, calibration), or at least a reference paper would be appreciated.**

Response:
References has been included and some additional information about uncertainty. Calibration was performed after the installation of the instrument at JPL-TMF and will be recalibrated during next summer.

Lines 196-203: *"Continuous surface ozone measurements have been performed at TMF since 2013 using the UV photometry technique (Huntzicker et al., 1979) with a UV photometric ozone analyzer (Model 49i from Thermo Fisher Scientific, US). The operation principle is based on the absorption of UV light at 254 nm by the ozone molecules (Shina et al., 2014). The instrument collects in-situ air samples at 2 meter above ground taken from an undisturbed forested environment adjacent to the lidar building. It provides ozone mixing ratio values at 1-minute time intervals with a lower detection limit of 1 ppbv. Uncertainty has been reported to be below 6% in previous studies (Shina et al., 2014)"*

**Comment:**
**- Line 241: Figure 3a is not restricted to the tropospheric part of the ozone profile. Either change this adjective in the sentence or the figure. The title of the paper and of the sub-section is clearly on tropospheric ozone variability. I suggest to plot Figure 3 only for altitudes up to 15 km (on average) and modify the X axis for linear concentrations on the tropospheric typical range [0-150 ppb]. In the seasonal part of the Figure 3, that will give the advantage to further see differences between the averaged seasonal vertical profiles. Is the spring profile that close to the summer profile ? From Figure 4, it seems that the spring profile should highlight higher ozone that the summer one.**

Response:
The paper is mainly focused on the tropospheric region. Nonetheless, ozone information in the UTLS region is very valuable and affects the ozone budget in the troposphere. Therefore we consider this information is relevant and should be maintained in the study. The sentence has been modified accordingly. A zoomed version of figure 3 has been included also in the manuscript to see more clearly the seasonal differences in the tropospheric part of the profiles. Larger values are observed in spring, but differences with summer are not very large.

**Comment:**
**- Lines 268-270: The spring-summer maximum is indeed a common characteristic of northern mid-latitudes over Europe and North-America. However, it seems that these JPL-TMF data as shown in Figure 4 shows a clear spring maximum (at least April-May). This seems actually consistent with what Zbinden et al, (2013) have shown over Los Angeles airport based on MOZAIC data. This paragraph deserves further clarification to highlight this "local" characteristic of a clear spring maximum (if confirmed by Figure 3 plotted with**

**a tropospheric X axis only). This special feature is likely the result of stronger influence from Asia in spring over the west coast of the US (Jaffe et al.2003; Parrish et al., 2004; Cooper et al., 2005; Neuman et al., 2012) as mentioned by Zbinden et al., (2013). Therefore, it is very surprising to read in the abstract that "No outstanding influence from Asia was identified". I am not convinced by such statement. Further comments below regarding subsection 3.4.**

**Response:**

In the paper by Zbinden et al., (2013) a strong maximum is observed in spring whereas values in summer are close to those observed in fall. At JPL-TMF, there is a maximum in spring but values are very similar to those observed in summer and clearly distinct from those in fall and winter (see attached figure and new version of Figure 3). Discrepancies between both datasets can be easily explained considering the different sampling periods (1994-2009 in Zbinden et al., (2013)) and the lower number of profiles (300 and <42 monthly profiles) used in the study by Zbinden et al., (2013).

[Figure]

Figure. a) Ozone mixing ratio climatological average (2000-2015) computed from the TMF lidar measurements (red curve). The cyan horizontal bars indicate the standard deviation at intervals of 1-km. The red dot at the bottom indicates the mean surface ozone mixing ratio (2013-2015) measured simultaneously with lidar. b) Seasonally-averaged ozone mixing ratio profiles for spring (MAM), summer (JJA), fall (SON) and winter (DJF).

**Comment:**

**- Lines 302-311: As far as I know, the common rule for indicator of the statistical significance of the trends with such procedure is to use p-Values lower than 0.05 for trends with a confidence level larger than 95% (and not p larger than 0.1 and confidence level larger than 90%). I recommend the authors to rewrite tables 2 and 3 applying these criteria. It turns out that only spring (in the upper troposphere) and winter (from 4 to 10 km) show significant seasonal trends. It would also be nice to give indication of error in % in first part of Table 3 as done in Table 2.**

Response:
A discussion using p-Values of 0.05 is also included to ease comparison with previous studies. Nonetheless, the criterion of p-Values lower than 0.1 is used in atmospheric sciences (e. g. Xia et al., 2008; Wilson et al., 2012) and we consider it is worthy to maintain it in the study. A confidence level larger than 90% is still quite significant and these trends should not be neglected. Tables and text has been modified accordingly. Error in % is now included in Table 3 as suggested by the reviewer.

Xia, X., T. F. Eck, B. N. Holben, G. Phillippe, and H. Chen (2008), Analysis of the weekly cycle of aerosol optical depth using AERONET and MODIS data, J. Geophys. Res., 113, D14217, doi:10.1029/2007JD009604.

Wilson, R. C., Fleming, Z. L., Monks, P. S., Clain, G., Henne, S., Konovalov, I. B., Szopa, S., and Menut, L.: Have primary emission reduction measures reduced ozone across Europe? An analysis of European rural background ozone trends 1996–2005, Atmos. Chem. Phys., 12, 437-454, doi:10.5194/acp-12-437-2012, 2012.

**Comment:**
**- Lines 322-328: This decrease in winter is the most surprising information of this paper. As far as I know, there is no other reference mentioning such significant decrease in winter in the region. In the recent review paper of Cooper et al., 2014 (and references therein including Cooper et al., (2012)), reported stations in California do show significant increase in winter. Therefore I recommend the authors to further argue and explain why such a different behavior at JPL-TMF station. This entire subsection misses further discussion on this trends analysis. See additional comments below regarding section 4.**

Response:
As previously stated, the analysis on the causes of the different trends is not straightforward and it is worthy an additional paper using complementary data (i.e. model, satellite, sounding data…). A possible cause for the differences between JPL-TMF and other stations such as those included in Cooper et al., (2014, 2012) might be related to the different sampling or the differences in the analyzed period. Values in the period 2011-2015, not included in the study by Cooper et al., (2012) seem to strongly contribute to the decreasing trend (see Figure 6). Excluding this years from the analysis and analyzing the same period as in Cooper et al., (2012), we still observe a negative trend at TMF (-0. 07 ppbv.year-1), but it is not statistically significant for this shorter period (p=0.83). These discrepancies between the different studies highlight the strong influence of sampling on the obtained results. Therefore, because of ozone large variability there is a need for long-term databases which can provide global coverage using

homogenous sampling in order to accurately characterized ozone trends in the troposphere as already pointed out in studies such as Lin et al., (2015b).

**Comment:**
**- Sub-section 3.4: This section is the weakest point of the draft paper. My major concern can be summarized as 2 questions: why (only) 8-days backward trajectories ? why (so many) 2-days of residence time over Asia ? These arbitrary choices need further arguments, sensitivity analysis and/or references for similar studies. From my knowledge, Cooper et al, (2010) used 15-days backward trajectories and concluded with a significant influence from Asia to the Western US region, especially in spring. To me, the length of the trajectories used here may be too short and the classification criteria is not adapted, to really assess the different influences authors are looking for. The two plots on Figure 8, bottom row, are too similar to trust in this methodology of classification. The criteria for air masses to be classified as ABL or AFT (at least 2 days over there) is too difficult to meet. What is the reason to impose 2 days over the continent to classify air parcels as "Asian" ? It is way too much. This is probably the reason why the Asian influence does not appear as strong as expected from previous studies. I guess that some (or most of) the air masses classified as Pacific (especially in Spring) would be classified Asian with a different criteria. Results and conclusions may be different with longer trajectories and different criteria for classification. For example, I would suggest to change the order of the sequential attributions: By default, air parcels that are not "stratosphere" would be classified as "Asian" unless the trajectories spend the entire period (8 days or longer) over the Ocean or over Central America. Maybe I'm wrong but I recommend the authors to revise this sub-section to make it convincing. Besides, an interesting information from this analysis of classified air masses would be to check if there is a tendency or anomalous behaviors from one year to another. If I understand well the title and the introduction, this subsection should help further understand the trend analysis. This is not the case so far.**

Response:

A sensitivity test was performed to determine the duration of the trajectories and the time residence chosen. Regarding the duration of the trajectories, 8 days was preferred after analyzing trajectories from 5 to 10 days of duration. It was observed that it was time enough to detect the different transport paths we were interested in and 8-day backward trajectories were preferred over longer duration to keep the uncertainty as low as possible, since it increases with the trajectories duration. For the residence time, two days were firstly chosen in order to guarantee that the air masses had enough time to interact with the different ozone sources (e.g. stratosphere, mixing layer above Asia).

However we agree with the reviewer that the chosen parameters were not the most appropriate and the sensitivity test was not exhaustive enough, obtaining misleading results. In this new version of the manuscript, we have performed a more in-depth sensitivity analysis to optimize the residence times over each region and the duration of the trajectories. For this sensitivity test we varied the residence time over each region from 6 to 288 hours (when appropriate) and the duration of the trajectories between 5 and 15 days. Resulting composite ozone profiles, the

number of trajectories and the air masses paths associated to each category were analyzed in detail to optimize the criteria for the classification.

Regarding the duration of the trajectories, this new test reveals that that the composite profiles are not statistically different when using different duration of the trajectories. The main conclusions regarding the ozone mixing ratio values associated to each region are still valid, independently of the trajectories duration (see figure below). However, the number of trajectories associated to each region (included in table 4) do vary significantly, especially for the Asian air masses. For trajectories duration below 10 days, the number of trajectories from Asia is slightly underestimated. A detailed analysis of the trajectories indicates that a value of 12 days for the trajectories significantly improves the results and most of the Asian air masses are correctly identified. Therefore, the trajectories duration has been established at 12 days for the new version of the manuscript. The residence times above each region has been optimized already considering these 12-day backward trajectories.

[Figure]

Figure: Composite profiles of the ozone mixing ratio associated with the different categories for each season and for different duration of the trajectories: 8 days (solid line), 10 days (dashed line), 12 days (dot-dash line) and 15 days (dotted line). Results are shown only when the number of samples for a given category was larger than 5% of the total number of samples.

In the case of the stratosphere, a residence time of only six hours already shows the influence of the stratosphere on the profiles, which clearly show increased ozone values. Results are very similar for residence times between 6 and 48 hours. Longer residence time leads to an underestimation of the stratospheric cases, since a large fraction of the trajectories descend to the troposphere after this time. The attached figure shows the average composite profiles obtained varying the residence time for the stratosphere between 6 and 48 hours and the associated standard deviation. As indicated by the low standard deviation, results are not highly dependent on the chosen residence time when selected within this range (6-48 hours). The selection of the

residence time in the stratosphere affects the profiles associated to the other regions, as observed in the figure. The largest standard deviations are associated to the Asian air masses, but variability is still quite low for most of the profiles.

[Figure]

Figure: Mean composite profiles of the ozone mixing ratio associated with the different categories for each season. Error bars are the standard deviation obtained when varying the residence time for the stratosphere between 6 and 48 hours. Residence times for the other categories are fixed (Central America: 96 h, ABL and AFT: 6 h, and Pacific Ocean: 276 h) and the trajectories duration is 12 days. Results are shown only when the number of samples for a given category was larger than 5% of the total number of samples.

In the case of Central America, the residence time has been increased to 96 hours to avoid the influence of additional sources. When lower values were used, air masses coming from Asia were included within the Central America category leading to an overestimation of the number of cases and influencing the ozone values. Larger values of the residence time, on the other hand, lead to an underestimation of the number of cases. Additionally, the Central America region has been slightly extended further north to 40°N. Based on the analysis of the obtained trajectories, this extended region is more adequate to group the trajectories associated to the North American monsoon circulation.

For the Asian air masses, 6 hours residence time is enough to observe the influence of the Asian air masses on the ozone composite profiles. With the new criteria established for the previous regions and duration of the trajectories of 12 days, almost no variation is observed in the ozone values for residence times above Asia between 6 and 48 h hours. However, residence times longer than 6 hours are more restrictive leading to an underestimation of the Asian number of cases. As indicated by the reviewer, residence times of 48 hours for this region with a total

duration of the trajectories of 8 days were too restrictive and the number of Asian cases was strongly underestimated. The manuscript has been modified according to the new results.

The Pacific region includes only those air masses with a residence time larger than 276 hours in the region. That way, influence from additional sources is avoided and this region can be considered as our background region.

[Figure]

Figure: Mean composite profiles of the ozone mixing ratio associated with the different categories (except for the stratosphere) for each season. Error bars are the standard deviation obtained when varying the residence time for the ABL, AFT, Central America and Pacific regions between 6 and 288 hours. Residence times for the stratosphere are fixed to 12 hours and the trajectories duration is 12 days. Results are shown only when the number of samples for a given category was larger than 5% of the total number of samples.

Trends associated to every region and every season were analyzed, but the number of data is significantly reduced when they are divided in subcategories so no statistically significant results were found, as previously stated. The number of cases associated to each category per year does not show any tendency or anomalous behavior that could explain the interannual variability observed in the profiles (see attached figure). Finally, no significant correlation is found between the number of cases of each category per year and the ozone values either. Even though no significant information on the interannual variability is obtained from the analysis, we think that the information about the different ozone source regions and how they influence the ozone profiles above the site is still quite valuable for the study.

[Figure]

Figure: Number of cases corresponding to every region per year at different altitude levels. The number of cases is normalized to the total number of measurements per year.

**Comment:**

**- Subsection 3.5: A figure showing an individual profiles (not averaged as in Fig.14) with double tropopause would be good to further explain this characteristic to the nonexpert reader. This section should make the distinction clearer between the wording "double tropopause" and "tropopause folds". It is not the same. Sentence line 442-443 and legend of Figure 12 are ambiguous. Indeed, Randel et a., (2007) should be cited as a reference paper for characteristics of double tropopauses.**

Response:

An additional figure with an example of tropopause fold has been added to figure 14 and the text has been modified accordingly.

Text has also been modified to make a clearer distinction between "double tropopause" and "tropopause fold". The following text has been added to the manuscript:

Lines 513-516: *"Double tropopauses are usually expected to result from tropopause folds in the layer between the two identified tropopauses. Therefore, a common method used in the literature*

*to identify tropopause folds is to detect the presence of double tropopauses based on temperature profiles (e.g. Chen et al., 2011)."*

The reference Randel et al., (2007) has been included where appropriate.

**Comment:**

**- Section 4: This section is a bit too long. It starts like a summary but then includes analysis and explanations that would be better placed before in the associated subsections. More importantly, some highlighted discussions are mentioned as in contradiction with other studies without any further discussions. This needs to be further argued. For example : - Line 477-484: Either the comparison with recent findings by Lin et al., (2015) and Neu et al., (2014) makes senses and the inconsistency raises important questions (i.e., what makes JPL-TMF station different and not representative of the general behavior ?), or the comparison is not possible (need to explain why ?) and such paragraph is simply to be removed.**

**Response:**

Section 4 has been removed and discussion is now included under the corresponding results in section 3.

The comparison with Lin et al., (2015) and Neu et al., (2014) has been removed since a simple interpretation is not possible due to the different characteristics of the datasets. Their results refer to a wider region and column-integrated data, whereas we have a single site with different temporal and vertical sampling. Therefore, a straightforward comparison is not likely and this sentence can lead to uncertain interpretations.

**- Lines 498-503: This negative trend in winter observed at JPL-TMF is surprising, in contradiction with most of the studies I'm aware of, and therefore deserves further investigations. According to Cooper et al., (2012) only one site in the Western US shows negative trend in winter, and only for the 95% percentile. This is different from what is presented in Table 2 in this study. "Decrease in background ozone during these months : : : " is definitely an opposite conclusion to that of Cooper et al., (2012) and of similar recent studies. As for previous comment, I recommend the authors to answer the question "why do JPL-TMF measurements highlight different behaviors ?"**

Response:
The sentence "Decrease in background ozone during these months…" has been removed.
As previously explained, differences between our results and previous studies might be due to differences in sampling and the different analyzed period compared to Cooper et al., (2012). Possible causes for the decreasing trend in winter have been investigated, but no significant results that can satisfactorily explain this trend were obtained. According to sections 3.4 and 3.5, an important source of ozone during winter would be the stratosphere, but no significant trends

for the ozone values associated to the stratospheric air masses were found in winter. No anomalous behavior or significant decrease in the number of stratospheric cases was observed during the analyzed period. The same results were obtained for the Asian air masses and the tropopause folds data during winter. It is worthy to note that the number of data is significantly reduced when they are subdivided in seasons and in the different categories associated to the five regions considered in the trajectories analysis, affecting the significance of the results. A combination of the lidar database with additional data (e.g. model, satellite, additional stations…) would be necessary to reach significant conclusions in this respect.

[Figure]

Figure: Number of cases corresponding to every region per year at different altitude levels in winter. The number of cases is normalized to the total number of measurements per year in winter.

**Comment:**
**- Line 513-517: Wouldn't it be good to have results from this extended analysis ?**

Response:
This study is undergoing and robust and reliable results are not yet available to be published. The sentence has been removed.

**Minor comments :**
**- Line 61 : "directly emitted" seems to me too ambiguous and may let the reader think that ozone is a primary pollutant. I suggest to replace by "transported down from the stratosphere".**

This part has been modified according to both reviewers' suggestions.

Lines 64-65: *"Tropospheric ozone is primarily formed as a secondary pollutant in chemicals reactions involving ozone precursors such as methane, CO, NOx, VOCs or PANs."*

**- Line 88-89 : I suggest to remove the end of the sentence starting at "which has not yet been : : :" because this not true, as written indeed in the following paragraph.**

The sentence has been rewritten:

Lines 96-99: *"In the last decades, efforts have been made in this respect and the number of tropospheric ozone measurements has considerably increased throughout the globe. However, it is still necessary to increase the current observation capabilities to characterize tropospheric ozone variability more accurately."*

**- Line 285-286: Do R=0.34 or 0.44 really show correlation?**

This paragraph has been removed considering both reviewers' comments.

**- Line 287-290: This is more interesting than the lines before and deserves a brief explanation. Why outliers have to be removed to confirm the correlation. Does this tell us something on specific process at the surface or at 4-6 km altitude? Is there different disconnected influences ? For a specific season?**

A first quick analysis reveals that outliers are mostly related to different ozone sources affecting the surface and the layer at 4-6 km. For example, stratospheric air going down to 4-6 km and not reaching the surface or local ozone production related to anthropogenic emissions near JPL-TMF site for those cases when the boundary layer reaches the station. However, the database available is not large enough to obtain significant and robust conclusions since surface data are available only since 2013. Especially, the number of cases is very low to try to identify any seasonal variation. According to both reviewers' comments on this section it has been removed from the manuscript.

**- Figure 5: I am wondering if the choice of the color sale is the most appropriate to highlight (real and significantly positive or negative) anomalies. What about choosing a color scale centered on 0 (same color for -10 to +10) ? It is difficult to check consistency with Figure 6. Is there any explanation for long-lasting anomalies in 2003-2007 ?**

Comparison between figure 6 and 7 is not straightforward since the first one shows monthly deviations from the climatological values in percentage and figure 6 is based on yearly median values expressed in ppbv for certain layers. We think that the use of the same color for positive and negative values suggested by the reviewer would mean a loss of information regarding the positive and negative anomalies.
Regarding the long-lasting positive anomalies in 2003-2007, we analyzed correlation with global circulation patterns such as ENSO or the QBO to identify the causes of the anomalies in 2003-2007, but no significant influence was observed. Additionally, a significant higher influence of any of the air masses from the five considered categories in Section 3.4 (Stratosphere, Central

America, ABL, AFT and Pacific) or the tropopause folds in Section 3.5 was not detected during this period.

**- Line 341: This reference is missing in the list.**

We apologize for the mistake. Reference has been added to the list.
Engström, A. and Magnusson, L.: Estimating trajectory uncertainties due to flow dependent errors in the atmospheric analysis, Atmos. Chem. Phys., 9, 8857–8867, doi:10.5194/acp-9-8857-2009,2009.

**- Line 412-414: This sentence needs to be accompanied by a reference to give argument that an excess of 15 ppbv is what is expected as lightning-induced enhancement of ozone. Is Cooper et al. (2009) as mentioned in section 4 (line 537) relevant for this ?**

The different between the ozone associated to the air masses classified as Central America and the Pacific Ocean is now 20 ppbv based on the new analysis in Section 3.4. We do not mean to attribute this ozone excess for the Central America air masses exclusively to lightning-induced enhancement of ozone. The sentence is meant to highlight that we have higher ozone mixing ratio values when air masses are coming from Central America. Cooper et al., (2009) shows an increased value of NOx based on models above JPL-TMF associated to lightning during the North American Monsoon but does not provide a corresponding ozone value. Lightning-induced NOx is an ozone precursor and the larger ozone values associated to Central American masses observed in our study seem to be the result of local ozone production associated to this enhanced NOx production. However, with the methodology used in this study an accurate quantification of the ozone associated to lightning is not possible. The use of chemical transport models would be required for that quantification. The possibility of a fraction of this ozone associated to different sources such as transport from Central America or even mixing with different sources such as the stratosphere cannot be ignored. Text has been modified to avoid confusion.

**- Line 429: MERRA acronym needs to be explained. Reference would be nice.**

Lines 516-518: *"The MERRA (Modern-Era Retrospective analysis for Research and Applications, Rienecker et al., 2011) reanalysis data (1-km vertical resolution, 1 x 1.25 degrees horizontal resolution) were used in this study to identify the presence of double tropopauses above the station."*

**- Reference list:**

**The following references are incomplete :**

**Ambrose et al., 2011; Cooper and Stohl, 2005; Lee and Akimoto, 1998; Monks, 2005; Petetin et al., 2015 (should check if ACP reference is available).**

**References :**
**Cooper, O. R., Stohl, A., Eckhardt, S., Parrish, D. D., Oltmans, S. J., Johnson, B. J., Nédélec P., Schmidlin, F. J., Newchurch, M. J., Kondo, Y., and Kita, K.: A springtime**

comparison of tropospheric ozone and transport pathways on the east and west coasts of the United States, J. Geophys. Res., 110, D05S90, doi: 10.1029/2004JD005183, 2005.

Jaffe, D., Price, H., Parrish, D., Goldstein, A., and Harris, J.: Increasing background ozone during spring on the west coast of North America, Geophys. Res. Lett., 30, 1613, doi:10.1029/2003GL017024, 2003.

Logan J.A., J. Staehelin, I. A. Megretskaia, J.-P. Cammas, V. Thouret, H. Claude, H. De Backer, M. Steinbacher, H. E. Scheel, R. Stübi, M. Fröhlich, and R. Derwent, Changes in ozone over Europe since 1990: analysis of ozone measurements from sondes, regular aircraft (MOZAIC) and alpine surface sites. J. Geophys. Res., D09301, doi:10.1029/2011JD016952, 2012.

Neuman, J. A., Trainer, M., Aikin, K. C., Angevine, W. M., Brioude, J., Brown, S. S., de Gouw, J. A., Dube, W. P., Flynn, J. H., Graus, M., Holloway, J. S., Lefer, B. L., Nedelec, P., Nowak, J. B., Parrish, D. D., Pollack, I. B., Roberts, J. M., Ryerson, T. B., Smit, H., Thouret, V., and Wagner, N. L.: Observations of ozone transport from the free troposphere to the Los Angeles basin, J. Geophys. Res., 117, D00V09, doi: 10.1029/2011JD016919, 2012.

Parrish, D., Dunlea, E. J., Atlas, E. L., Schauffler, S., Donnelly, S., Stroud, V., Goldstein, A. H., Millet, D. B., McKay, M., Jaffe, D. A., Price, H. U., Hess, P. G., Flocke, F., and Roberts, J. M.: Changes in the photochemical environment of the temperate North Pacific troposphere in response to increased Asian emissions, J. Geophys. Res., 109, D23S18, doi: 10.1029/2004JD004978, 2004.

Randel, W. J., D. J. Seidel, and L. L. Pan, Observational characteristics of double tropopauses, J. Geophys. Res., 112, D07309, doi:10.1029/2006JD007904, 2007.

Zbinden R.M., V. Thouret, P. Ricaud, F. Carminati, J.-P. Cammas, and P. Nédélec, Climatology of pure tropospheric profiles and column contents of ozone and carbon monoxide using MOZAIC in the mid-northern latitudes (24_ N to 50_ N) from 1994 to 2009,, Atmos. Chem. Phys., 13, 12363-12388, doi:10.5194/acp-13-12363-2013, 2013.

References has been modified.

---

## Author Comment (AC2)

**General Comments:**
**This paper is a very useful contribution to the rather limited literature of the vertical distribution of ozone over the USA west coast. The analytical techniques employing trajectory analysis, stratospheric-tropospheric folding dynamical structures, time series and variability analysis, and attribution analysis all contribute to the value of this work. The major shortcomings concern the choices for trajectory/attribution parameters (primarily time scales) and the absence of a concise conclusion section. Minor shortcomings concern the details of the trend analysis and some inconsistencies in the attribution. This paper should be accepted after these issues are successfully addressed.**

Response:

First of all, we would like to thank the reviewer for the comments and suggestions which have helped to improve the quality of the manuscript. Especially, section 3.4 is now much improved and more significant results were found. Comments and suggestions have been really helpful to improve the section and the quality of the manuscript.

The trajectories attribution section has been modified considering the suggestions and comments from both reviewers. An in-depth sensitivity analysis to optimize the residence times over each region and the duration of the trajectories has been performed and new criteria have been established. Additional details on the selection of the parameters and the duration of the trajectories are now included in the manuscript. More information is provided next.

For the sensitivity test we varied the residence time over each region from 6 to 288 hours (when appropriate) and the duration of the trajectories between 5 and 15 days. Resulting composite ozone profiles, the number of trajectories and the air masses paths associated to each category were analyzed in detail to optimize the criteria for the classification.

Regarding the duration of the trajectories, the sensitivity test reveals that that the composite profiles are not statistically different when using different duration of the trajectories. The main conclusions regarding the ozone mixing ratio values associated to each region are still valid, independently of the trajectories duration (see figure below). However, the number of trajectories associated to each region (included in table 4 in the manuscript) do vary significantly, especially for the Asian and Pacific air masses. For trajectories duration below 10 days, the number of trajectories from Asia is slightly underestimated. A detailed analysis of the trajectories indicates that a value of 12 days for the trajectories significantly improves the results and most of the Asian air masses are correctly identified. Therefore, the trajectories duration has been established

at 12 days for the new version of the manuscript. The residence times above each region has been optimized already considering these 12-day backward trajectories.

[Figure]

Figure: Composite profiles of the ozone mixing ratio associated with the different categories for each season and for different duration of the trajectories: 8 days (solid line), 10 days (dashed line), 12 days (dot-dash line) and 15 days (dotted line). Results are shown only when the number of samples for a given category was larger than 5% of the total number of samples.

In the case of the stratosphere, a residence time of only six hours already shows the influence of the stratosphere on the profiles, which clearly show increased ozone values. Results are very similar for residence times between 6 and 48 hours. Longer residence time leads to an underestimation of the stratospheric cases, since a large fraction of the trajectories descend to the troposphere after this time. The attached figure shows the average composite profiles obtained varying the residence time for the stratosphere between 6 and 48 hours and the associated standard deviation. As indicated by the low standard deviation, results are not highly dependent on the chosen residence time when selected within this range (6-48 hours). The selection of the residence time in the stratosphere affects the profiles associated to the other regions, as observed in the figure. The largest standard deviations are associated to the Asian air masses, but variability is still quite low for most of the profiles.

[Figure]

Figure: Mean composite profiles of the ozone mixing ratio associated with the different categories for each season. Error bars are the standard deviation obtained when varying the residence time for the stratosphere between 6 and 48 hours. Residence times for the other categories are fixed (Central America: 96 h, ABL and AFT: 6 h, and Pacific Ocean: 276 h) and the trajectories duration is 12 days. Results are shown only when the number of samples for a given category was larger than 5% of the total number of samples.

In the case of Central America, the residence time has been increased to 96 hours to avoid the influence of additional sources. When lower values were used, air masses coming from Asia were included within the Central America category leading to an overestimation of the number of cases and influencing the ozone values. Larger values of the residence time, on the other hand, lead to an underestimation of the number of cases. Additionally, the Central America region has been slightly extended further north to 40°N. Based on the analysis of the obtained trajectories, this extended region is more adequate to group the trajectories associated to the North American monsoon circulation.

For the Asian air masses, 6 hours residence time is enough to observe the influence of the Asian air masses on the ozone composite profiles. With the new criteria established for the previous regions and duration of the trajectories of 12 days, almost no variation is observed in the ozone values for residence times above Asia between 6 and 48 h hours. However, residence times longer than 6 hours are more restrictive leading to an underestimation of the Asian number of cases. Residence times of 48 hours for this region with a total duration of the trajectories of 8 days were too restrictive and the number of Asian cases was strongly underestimated. The manuscript has been modified according to the new results.

The Pacific region includes only those air masses with a residence time larger than 276 hours in the region. That way, influence from additional sources is avoided and this region can be considered as our background region.

[Figure]

Figure: Mean composite profiles of the ozone mixing ratio associated with the different categories (except for the stratosphere) for each season. Error bars are the standard deviation obtained when varying the residence time for the ABL, AFT, Central America and Pacific regions between 6 and 288 hours. Residence times for the stratosphere are fixed to 12 hours and the trajectories duration is 12 days. Results are shown only when the number of samples for a given category was larger than 5% of the total number of samples.

A conclusion sections was already added to the manuscript published in ACPD. Please note that the version published and the version attached by the reviewer as a supplement are different. The conclusions section has now been modified in the new version of the manuscript according to the new results obtained in Section 3.4.

**Specific Comments: L36: 'No outstanding influence from Asia was identified'. This absence of Asian influence is strongly dependent on the somewhat arbitrary selection of trajectory time-scale parameters. This conclusion is also somewhat inconsistent with the early spring maximum in figure 4. Consider additional analyses to resolve this discrepancy by providing compelling evidence to support your finding.**

Response:

We agree with the reviewers that the chosen parameters in Section 3.4 were not the most appropriate and the sensitivity test performed was not exhaustive enough, obtaining misleading results. As previously explained, a new analysis has been performed based on a more exhaustive sensitivity analysis to optimize the residence times over each region and the duration of the trajectories. The new criteria used in the back-trajectories analysis section reveal now the influence of air masses coming from Asia on the ozone profiles measured at JPL-TMF and results are in agreement with previous studies. The manuscript has been modified accordingly.

**L44: 'Tropospheric ozone can be directly emitted to the troposphere, ': Direct emissions (separate from STE injections) are a very small fraction of tropospheric ozone sources. Suggest you omit this sentence.**

Response:
Sentence has been removed.

**L273: Removing data +/- 1 sd for a correlation calculation is not a legitimate approach. That process will remove approximately 1/3 of the data and will certainly enhance the correlation between the remaining data, but one cannot justify removing that many data and one would certainly not call all those data 'outliers'.**

Response:
This paragraph has been omitted from the manuscript considering both reviewers' concerns.

**L296: Suggest you use p-values of 0.05 to be consistent with the 95% statistics used elsewhere.**

Response:
A discussion using p-Values of 0.05 is also included to be consistent and ease comparison with previous studies. Nonetheless, the criterion of p-Values lower than 0.1 is also used in atmospheric sciences (e. g. Xia et al., 2008; Wilson et al., 2012) and we consider it is worthy to maintain it in the study. A confidence level larger than 90% is still quite significant and these trends should not be neglected. Tables and text has been modified accordingly.

Xia, X., T. F. Eck, B. N. Holben, G. Phillippe, and H. Chen (2008), Analysis of the weekly cycle of aerosol optical depth using AERONET and MODIS data, J. Geophys. Res., 113, D14217, doi:10.1029/2007JD009604.

Wilson, R. C., Fleming, Z. L., Monks, P. S., Clain, G., Henne, S., Konovalov, I. B., Szopa, S., and Menut, L.: Have primary emission reduction measures reduced ozone across Europe? An analysis of European rural background ozone trends 1996–2005, Atmos. Chem. Phys., 12, 437-454, doi:10.5194/acp-12-437-2012, 2012.

**Section 4: The summary should be expressed in the Abstract. No need for another summary here. The more discussions should be moved to the section under discussion or a new section heading inserted. The paper needs a short 'Conclusions' section (not summary or discussion). The conclusions should be succinct and describe the main take-home points derived from the paper.**

Response:
The discussions have been inserted in Section 3 with the corresponding results as suggested by Reviewer 1. Therefore the discussion section is no longer included in the manuscript.

**Technical corrections: See attached .docx for suggested tracked changes.**
**Please also note the supplement to this comment:**

**http://www.atmos-chem-phys-discuss.net/acp-2016-70/acp-2016-70-RC2-supplement.pdf**

Response:
Technical corrections suggested by the reviewer have been addressed in the new version of the manuscript.

Regarding the comment on Line 500 (The wintertime negative results require an explanation), additional details has been added to explain the discrepancies with previous studies. A possible cause for the differences between JPL-TMF and other stations such as those included in Cooper et al., (2014, 2012) might be related to the different sampling or the differences in the analyzed period. Values in the period 2011-2015, not included in the study by Cooper et al., (2012) seem to strongly contribute to the decreasing trend (see Figure 6). Excluding this years from the analysis and analyzing the same period as in Cooper et al., (2012), we still observe a negative trend at TMF (-0. 07 ppbv.year$^{-1}$), but it is not statistically significant for this shorter period (p=0.83). These discrepancies between the different studies highlight the strong influence of sampling on the obtained results, as already suggested in Lin et al., (2015b). Possible causes for the decreasing trend in winter have been investigated, but no significant results that can satisfactorily explain this trend were obtained. According to sections 3.4 and 3.5, an important source of ozone during winter would be the stratosphere, but no significant trends for the ozone values associated to the stratospheric air masses were found in winter. No anomalous behavior or significant decrease in the number of stratospheric cases was observed either during the analyzed period. The same results were obtained for the Asian air masses and the tropopause folds data during winter. It is worthy to note that the number of data is significantly reduced when they are subdivided in seasons and in the different categories associated to the five regions considered in the trajectories analysis, affecting the significance of the results. A combination of the lidar database with additional data (e.g. model, satellite, additional stations…) would be necessary to reach significant conclusions in this respect.